# Biogeochemical modeling of $CO_2$ and $CH_4$ production in anoxic Arctic soil microcosms

Guoping Tang[1], Jianqiu Zheng[2], Xiaofeng Xu[3], Ziming Yang[1], David E. Graham[2,4], Baohua Gu[1], Scott Painter[1,4], and Peter E. Thornton[1,4]

[1]Environmental Sciences Division, Oak Ridge National Laboratory, Oak Ridge TN, 37831 USA
[2]Biosciences Sciences Division, Oak Ridge National Laboratory, Oak Ridge TN, 37831 USA
[3]Biology Department, San Diego State University, San Diego, CA, 92182 USA
[4]Climate Change Science Institute, Oak Ridge National Laboratory, Oak Ridge TN, 37831 USA

*Correspondence to*: Guoping Tang (guopingtangva@gmail.com)

**Abstract.** Soil organic carbon turnover to $CO_2$ and $CH_4$ is sensitive to soil redox potential and pH conditions. However, land surface models do not consider redox and pH in the aqueous phase explicitly, thereby limiting their use for making predictions in anoxic environments. Using recent data from incubations of Arctic soils, we extend the Community Land Model Carbon Nitrogen (CLM-CN) decomposition cascade to include simple organic substrate turnover, fermentation, Fe(III) reduction, and methanogenesis reactions, and assess the efficacy of various temperature and pH response functions. Incorporating the Windermere Humic Aqueous Model (WHAM) enables us to approximately describe the observed pH evolution without additional parameterization. Although Fe(III) reduction is normally assumed to compete with methanogenesis, the model predicts that Fe(III) reduction raises the pH from acidic to neutral, thereby reducing environmental stress to methanogens and accelerating methane production when substrates are not limiting. The equilibrium speciation predicts a substantial increase in $CO_2$ solubility as pH increases, and taking into account $CO_2$ adsorption to surface sites of metal oxides further decreases the predicted headspace gas-phase fraction at low pH. Without adequate representation of these speciation reactions, and the impacts of pH temperature, and pressure, the $CO_2$ production from closed microcosms can be substantially underestimated based on headspace $CO_2$ measurements only. Our results demonstrate the efficacy of geochemical models for simulating soil biogeochemistry and provide predictive understanding and mechanistic representations that can be incorporated into land surface models to improve climate predictions.

**Key words.** anaerobic incubation, Fe(III) reduction, methanogenesis, pH, temperature response

## 1 Introduction

Global warming is expected to accelerate permafrost thaw, which may trigger the release of the large amount of frozen soil organic matter (SOM) stored in the Arctic as carbon dioxide ($CO_2$) and methane ($CH_4$) into the atmosphere, possibly forming a positive feedback to climate change (Treat et al., 2015; Knoblauch et al., 2013; Elberling et al., 2013). Permafrost thawing leads to significant changes in soil water saturation, creating favorable conditions for anaerobic respiration and methanogenesis (Lawrence et al., 2015).

Current biogeochemical models predominantly represent SOM decomposition under aerobic conditions (Manzoni and Porporato, 2009). They are modified for use under anaerobic conditions. For example, the Community Land Model Carbon Nitrogen (CLM-CN) decomposition cascade is used to implicitly represent anaerobic decomposition with a moisture response function that approaches unity at saturation and an oxygen scalar that has a large unresolved uncertainty (Oleson et al., 2013). In a recent permafrost carbon climate feedback modeling study, the carbon release rate from permafrost soils after thawing under aerobic conditions was assumed to be 3.4 times higher than the release rate under anaerobic conditions (Koven et al., 2015; Schädel et al., 2016). However, in incubations with soils from Alaska and Siberia, carbon release under aerobic conditions was 3.9–10 times greater than under anaerobic conditions (Lee et al., 2012), and $CO_2$ production appeared ceased at late times in anaerobic microcosms (Xu et al., 2015; Roy Chowdhury et al., 2015), indicating that these existing models do not adequately represent the anaerobic processes for accurate prediction of SOM turnover and heterotrophic respiration.

In addition, it is important to accurately represent methanogenesis in the context of competing anaerobic processes because $CH_4$ has a 100-year global warming potential that is about 26 times greater than $CO_2$ (Forster et al., 2007; IPCC, 2013), an atmospheric residence time of approximately 10 years (IPCC, 2013), and methanogenesis rate can be high under favorable conditions,. Methanogenesis is carried out by a group of strictly anaerobic Archaea. The free energy of methanogenesis reactions is less favorable than the reduction of $O_2$, $NO_3^-$, Mn (IV), Fe(III), and $SO_4^{2-}$ along the redox ladder (Conrad, 1996; Bethke et al., 2011). The accumulation of $CH_4$ has been widely observed to lag behind $CO_2$ for periods ranging from days to years in incubations (Knoblauch et al., 2013; Roy Chowdhury et al., 2015; Cui et al., 2015; Hoj et al., 2007; Fey et al., 2004; Jerman et al., 2009; Tang et al., 2013c). The implication is that a first order representation (including constant $CO_2$ $CH_4$ ratio parameterization) normally overpredicts $CH_4$ production rate before methanogenosis initiation and underpredicts $CH_4$ production rate afterwards, and the uncertain lag time introduces large bias in $CH_4$ production prediction.

Besides temperature (Fey and Conrad, 2003; Hoj et al., 2007; Jerman et al., 2009; Cui et al., 2015) and initial methanogen abundance (Conrad, 1996; Knoblauch et al., 2013), the wide range of redox buffers provided by the alternative electron acceptors is likely a cause of the wide range of observed lag times (Estop-Aragonés and Blodau, 2012; Fey et al., 2004;

Jerman et al., 2009; Yao et al., 1999; Conrad, 1996; Knorr and Blodau, 2009). As a result, the ratio of $CH_4$ to $CO_2$ ranges from 0.00001 to 0.5 (Wania et al., 2010; Drake et al., 2009; Bridgham et al., 2013), highlighting limitation of the $CH_4$ $CO_2$ ratio approach. Nevertheless, some land surface models (LSM) parameterize methanogenesis as a fraction of carbon mineralization (Wania et al., 2013; Oleson et al., 2013; Koven et al., 2015; Cheng et al., 2013). While methanogenesis is

5    explicitly represented in some models (Xu et al., 2015; Grant, 1998) and the reduction of alternative electron acceptors is explicitly represented in others (Fumoto et al., 2008; Segers and Kengen, 1998; Van Bodegom et al., 2001; van Bodegom et al., 2000), these models do not have an aqueous phase that is essential for explicit biogeochemical calculations, e.g., pH, Eh, and thermodynamic calculations. Because methanogenesis is sensitive to redox conditions, the lack of explicit biogeochemical representation of the redox processes contributes to the prediction uncertainty of $CH_4$ emission.

Anaerobic bacteria and Archaea usually depend on simple substrates such as sugars, alcohols, organic acids, and $H_2$ as carbon and energy sources that are rarely simulated in ecosystem models (Manzoni and Porporato, 2009; Xu et al, 2015). Instead, they are typically lumped together as dissolved organic matter (DOM) or low-molecular-weight organic carbon (LMWOC) (e.g., Tian et al., 2010). The abundance and importance of DOM and LMWOC in SOM turnover in the Arctic

soils are becoming increasingly recognized (Hodgkins et al., 2014). The DOM concentration in water flowing from collapsing permafrost (thermokarsts) on the North Slope of Alaska ranges from 0.2−8 mM, with biodegradable (degrade in 40 d) DOM accounting for 10-60 % (Abbott et al., 2014; Arnosti, 1998, 2000; Arnosti et al., 1998). Ancient LMWOC was found to fuel rapid $CO_2$ production upon thawing (Drake et al., 2015). On the other hand, new SOM consists of mostly macromolecules of plant and microbial residues such as carbohydrates (polysaccharides, including cellulose, hemicellulose,

etc.), lipids, nucleic acids, and proteins (Kögel-Knabner, 2002). While conceptual models and measurements connecting SOM with LMWOC have long existed (Drake et al., 2009; Tveit et al., 2013; Tveit et al., 2015; Bridgham et al., 2013), the hydrolysis and fermentation reactions have been poorly represented and quantified in the Arctic as well as temperate and tropical soils. Among over 250 SOM decomposition models that have been developed in the past 80 years (Manzoni and Porporato, 2009), only a few models explicitly simulate simple substrates (Xu et al., 2016b). Either a simple carbon pool

(Cao et al., 1995; Cao et al., 1998; Kettunen, 2003) or a DOM pool (Tian et al., 2010; Xu and Tian, 2012) has been assumed for methanogenesis. The production of acetate and $H_2$ has been parameterized as a function of carbon mineralization (Van Bodegom et al., 2001; van Bodegom et al., 2000; Grant, 1998; Xu et al., 2015). It is not surprising that $CH_4$ production prediction is sensitive to simple substrate production (Kettunen, 2003; Weedon et al., 2013). While detailed SOM decomposition models include depolymerization to produce monomers under aerobic conditions (Riley et al., 2014),

production and consumption of simple measurable substrates, such as acetate, $H_2$, formate, etc., are not explicitly represented under anaerobic conditions.

In addition to electron acceptors and substrates, SOM turnover is also sensitive to soil pH. Most methanogens grow over a relatively narrow pH range (6-8), while some adapt to acidic or basic environments (Garcia et al., 2000; Van Kessel and

Russell, 1996; Wang et al., 1993; Sowers et al., 1984; Rivkina et al., 2007; Hao et al., 2012; Kotsyurbenko et al., 2004; Kotsyurbenko et al., 2007). Soil pH can change 1-2 logarithmic units in laboratory incubations (Xu et al., 2015; Roy Chowdhury et al., 2015; Peters and Conrad, 1996; Drake et al., 2015) and it can vary significantly through the soil profile and along topographic and vegetation gradients in the field (Cao et al., 1995; Van Bodegom et al., 2001; Lipson et al., 2013b). The pH feedback on methanogenesis could be up to 30% (Xu et al., 2015). However, soil pH is often fixed in LSMs (Oleson et al., 2013; Tian et al., 2010). pH is calculated using soil acidity and soil buffer capacity (Van Bodegom et al., 2001) or as a function of acetate concentration (Xu et al., 2015). It is desirable to use a geochemical model to describe pH evolution mechanistically. The pH response functions (reaction rate adjustment factor as a function of pH) in LSMs are empirical and vary substantially (Xu et al., 2016b). Assessing the efficacy of these functions is needed to better represent pH impacts on carbon mineralization and methanogenesis.

Temperature is another critical factor controlling SOM turnover to $CO_2$ and $CH_4$. The reported $Q_{10}$ values for methanogen temperature response vary from 1.5 to 4 (Xu et al., 2016b). Methanogenesis has been widely observed to diminish when the temperature decreases toward 0 °C (Dunfield et al., 1993; Fey et al., 2004; Hoj et al., 2007; Sowers et al., 1984), predicting little $CH_4$ production from the surface layers of frozen Arctic soils. However, recent observations suggest that $CH_4$ emissions during the winter season account for $\geq 50$ % of the annual emission in the Arctic (Zona et al., 2016). The cold season $CH_4$ production is among the most uncertain processes for predicting seasonal $CH_4$ cycle in northern wetlands (Xu et al., 2016a). The temperature response functions (reaction rate adjustment factor as a function of temperature) need to be assessed as well.

Overall, anaerobic SOM turnover is controlled by the hydrolysis of the macromolecules to produce simple substrates and the sequential microbial reduction of electron acceptors along the redox ladder. Because SOM turnover and $CO_2$ and $CH_4$ productions are sensitive to redox potential, pH, and temperature, it is desirable to simulate the redox and pH explicitly with geochemical models. With the accumulation of new data on metabolic intermediates, electron acceptors, greenhouse gases, and pH from incubations with Arctic soils at various temperatures (Drake et al., 2015; Herndon et al., 2015a; Herndon et al., 2015b; Yang et al., 2016; Mann et al., 2015), our objectives are to integrate these new data into geochemical models to (1) extend the CLM-CN decomposition cascade to include simple substrates such as sugars and organic acids and add Fe(III) reduction and methanogenesis processes, (2) account for gas, aqueous, and adsorbed phase speciation, (3) describe pH mechanistically, and (4) assess the existing temperature and pH response functions. Unlike previous LSMs, we simulate speciation of $CO_2$ and $CH_4$ in the gas, aqueous, and solid phases, and represent sugars, organic acids, Fe(II), Fe(III), Fe reducers, and methanogens, and account for both thermodynamic and kinetic control. Our results provide predictive understanding and mechanistic representations that can be incorporated in LSMs, e.g., CLM-PFLOTRAN (Tang et al., 2016), to improve climate model predictions.

The carbon cycle involves coupled hydrological, geochemical, and biological processes interacting from molecular to global scales. The implicit empirical first order approach used in existing LSMs limits our understanding of the land atmosphere interaction and is a source of prediction uncertainty. To improve our understanding and reduce prediction uncertainty, we attempt to use relatively more explicit mechanistic representations developed in the reactive transport model literature (Tang et al., 2016). Even though explicit representation does not necessarily improve the match between the predictions and observations over well-tuned existing models immediately (e.g., Wieder et al. 2015; Steven et al. 2006), our approach provides a systematic means to incorporate on-going process-rich investigations to improve mechanistic representations in LSMs across scales. As a preliminary study, we constrain our scope to extending CLM-CN with minimum revision to describe anaerobic $CO_2$ and $CH_4$ production from several recent microcosm studies in this work. We discuss next steps briefly in the results and discussion section.

## 2 Materials and methods

We extend the CLM-CN decomposition cascade (Thornton and Rosenbloom, 2005) by adding reactions for hydrolysis to produce sugars, fermentation to produce organic acids and $H_2$ (Grant, 1998; Xu et al., 2015), Fe(III) reduction, and methanogenesis reactions (Tang et al., 2013c). We add the Windermere Humic Aqueous Model (WHAM) (Tipping, 1994) to simulate the pH buffer by SOM. Recent microcosm data (Herndon et al., 2015a; Roy Chowdhury et al., 2015) are used to assess these representations. While nitrogen (ammonium and nitrate) concentrations can affect carbon mineralization (Lavoie et al., 2011), we do not account for this effect because of a lack of nitrogen measurements from these experiments.

### 2.1 Soil incubation experiment data

The materials, experimental procedures and results for the microcosm tests were reported previously (Herndon et al., 2015a; Roy Chowdhury et al., 2015). Briefly, three soil cores were taken from center, ridge, and trough locations in a low-center polygon (a typical arctic geographic feature in the low lands with soils surround by ice wedges, see cited references for more information) in the wet tundra of the Barrow Environmental Observatory in Alaska. Soil samples from the organic and mineral horizons of the three cores were analyzed for gravimetric water content, pH, Fe(II), water-extractable organic carbon (WEOC), organic acids, and total organic carbon content (TOTC). For each horizon and location, about 15 g of homogenized wet soil was placed into a 60 ml sterile serum bottle, which was sealed and flushed with pure $N_2$ gas. The microcosms were incubated at -2, 4, and 8 °C for about 2 months to mimic thawing during the summer season at the site. The headspace $CO_2$ and $CH_4$ were sampled and analyzed by gas chromatography. Separate microcosms with 20 g of the homogenized soils were incubated to analyze for pH, Fe(II), water extractable organic carbon, and organic acids. Additional soil characterization is available elsewhere (Bockheim et al., 2001; Lipson et al., 2010; Lipson et al., 2013b).

## 2.2 Model developments

### 2.2.1 SOM decomposition

The SOM in the Arctic soils was characterized using high-resolution mass spectroscopy (Herndon et al., 2015a; Mann et al., 2015; Hodgkins et al., 2014). However, these characterizations were insufficient to partition SOM into many chemically distinct organic pools (Riley et al., 2014; Kögel-Knabner, 2002). Therefore, we extend the CLM-CN decomposition cascade to produce intermediate metabolites (Fig. 1). To limit the number of new pools, we lump reducing sugars, alcohols, etc. (Yang et al., 2016; Kotsyurbenko et al., 1993; Glissmann and Conrad, 2002; Tveit et al., 2015) into a LabileDOC pool, and the organic acids, such as formate, acetate, propionate, and butyrate, etc. (Herndon et al., 2015a; Kotsyurbenko et al., 1993; Peters and Conrad, 1996; Tveit et al., 2015) into an organic acid pool (Ac) (Xu et al., 2015; Grant, 1998). Assuming that the LabileDOC turns over in 20 h as the Lit1 pool in CLM-CN (Thornton and Rosenbloom, 2005) or glucose fermentation (Rittmann and McCarty, 2001), we split the original respiration factor into a direct and an indirect fraction, with the indirect fraction $s_{labile}$ to produce LabileDOC, which respires through the anaerobic pathway (Fig. 1) to $CO_2$ or $CH_4$, and the direct respiration fraction (1- $s_{labile}$) respires directly to $CO_2$. We estimate $s_{labile}$ by comparing the predictions with the observations in this work. The fermentation reaction is (Xu et al., 2015; Grant, 1998; van Bodegom and Scholten, 2001; Madigan, 2012)

$$C_6H_{12}O_6 + 4\ H_2O => 2\ CH_3COO^- + 2\ HCO_3^- + 4\ H^+ + 4\ H_2, \tag{R1}$$

which lowers the pH and further respires $s_{labile}/3$ of SOM into $CO_2$.

### 2.2.2 Fe(III) reduction, methanogenesis, and biomass decay

Because Fe(III) reduction contributes 40-45 % of the ecosystem respiration in some Arctic sites (Lipson et al., 2013b) and $NO_3^-$ and $SO_4^{2-}$ concentrations are typically low in the experiments, we add Fe(III) reduction reactions to represent the reduction of alternative electron acceptors to $O_2$. We use the microbial reactions formed by combining electron donor (oxidation) half reactions, electron acceptor (reduction) half reactions, and cell synthesis reactions following bioenergetics (Rittmann and McCarty, 2001). Specifically, the Fe(III) reduction reactions are (Istok et al., 2010)

$$2.1\ H_2O + NH_4^+ + 150.2\ Fe^{3+} + 21.3\ CH_3COO^- => C_5H_7O_2N + 150.2\ Fe^{2+} + 167.4\ H^+ + 37.5\ HCO_3^-, \tag{R2}$$

$$5\ HCO^{3-} + NH_4^+ + 114.8\ Fe^{3+} + 57.4\ H_2 => C_5H_7O_2N + 114.8\ Fe^{2+} + 110.8\ H^+ + 13\ H_2O, \tag{R3}$$

where, $C_5H_7O_2N$ represents microbial (iron reducer) mass, and $NH_4^+$ is assumed not to be limiting (at 1 μM). These two reactions result in dissolution of ferric oxides, for example, $Fe(OH)_{3a}$, to release $OH^-$ to increase pH. The rate is

$$\frac{dx}{dt} = k_{max} x \frac{k_{surf}}{k_{surf} + x/m_{surf,avail}} \frac{m_D}{k_D + m_D} f(G), \tag{1}$$

where $k_{max}$ is the kinetic rate constant; $x$ is concentration of biomass; $m_{surf,avail}$ is the microbially-available surface sites taken as the $Fe(OH)_{3a}$ surface sites Hfo associated with $H^+$, i.e., $m_{surf,avail} = m_{Hfo\_wOH} + m_{Hfo\_sOH}$ in moles per liter pore fluid; $k_{surf}$ accounts for the impact of $x/m_{surf,avail}$, which represents the interaction of biomass with available Fe(III) sites

on the surface; $m_D$ and $k_D$ are the concentration and half saturation of the electron donors (acetate or $H_2$); and $f(G)$ is a thermodynamic factor that goes to zero when the reaction is thermodynamically unfavourable (Jin and Roden, 2011).

The methanogenesis reactions are (Istok et al., 2010)

$$1.5\ H^+ + 98.2\ H_2O + NH_4^+ + 103.7\ CH_3COO^- => C_5H_7O_2N + 101.2\ HCO_3^- + 101.2\ CH_4, \tag{R4}$$

$$84.9\ H^+ + NH_4^+ + 85.9\ HCO_3^- + 333.5\ H_2 => C_5H_7O_2N + 255.6\ H_2O + 80.9\ CH_4. \tag{R5}$$

These two reactions consume protons to increase pH. The rate is

$$\frac{dx}{dt} = k_{max} x \frac{m_D}{k_D + m_D} f(G). \tag{2}$$

We use one pool FeRB for the iron reducers and separate the methanogens into MeGA and MeGH pools for acetoclastic and hydrogenotrophic methanogens (Fig. 1). The biomass decay reaction for FeRB, MeGA, and MeGH is

$$0.2\ C_5H_7O_2N => 0.1\ SOM1 + 0.2\ SOM2 + 0.25\ SOM3 + 0.45\ SOM4 + 0.1185\ NH_4^+ + \ldots \tag{R6}$$

Like the SOM pools, the rate is first order.

In this model, iron reducers and methanogens interact in different ways under various conditions. When the electron donors (acetate and $H_2$) are abundant, iron reducers grow faster than methanogens when Fe(III) is not limiting (depending on the $Fe(OH)_{3a}$ surface sites and iron reducers population), i.e., iron reducers have a short doubling time than methanogens. When the electron donors are limiting, iron reducers are expected to outcompete methanogens, depending on the half saturation (substrate affinity) values. The model also accounts for the thermodynamics. However, it does not account for possible different responses to temperatures and pH for iron reducers and methanogens.

### 2.2.3 pH

The soil pH is typically buffered by carbonates, clay minerals, metal oxides, and organic matter (Tipping, 1994; Tang et al., 2013a). The Windermere Humic Aqueous Model (WHAM) is used to approximate SOM as humic acid and fulvic acid, with a number of monodentate and bidentate binding sites for protons, to describe the pH buffering due to SOM (Tipping, 1994). The surface complexation model for ferrihydrate is used to describe the sorption of carbonate and proton to metal oxides (Dzombak and Morel, 1990). Additional aqueous speciation reactions are also included in the reaction database available as supplemental information (also publicly available at https://github.com/t6g/bgcs).

### 2.2.4 pH and temperature response functions

We use the CLM4Me pH response function (Riley et al., 2011; Meng et al., 2012)

$$log_{10} f(pH) = -0.2235 pH^2 + 2.7727 pH - 8.6 \tag{3}$$

and the CLM-CN temperature response function (Thornton and Rosenbloom, 2005; Lloyd and Taylor, 1994)

$$\ln f(T) = 308.56 \left( \frac{1}{71.02} - \frac{1}{T - 227.13} \right). \tag{4}$$

The pH response functions used in DLEM (Tian et al., 2010) and TEM (Raich et al., 1991) and a few other models (Cao et al., 1995; Xu et al., 2015), as described in Appendix 1, and the CENTURY temperature response function, the $Q_{10}$ equation, Arrhenius equation, and Ratkowsky equation, which are described in Appendix 2, are used for comparison.

## 2.3 Implementation, parameterization, and initialization

### 2.3.1 Implementation

To calculate the speciation of $CO_2$, $CH_4$, $H_2$, Fe, etc. among gas, aqueous, and solid phases under various temperature, pH, and pressure conditions and explicitly describe pH and redox buffer, we employ the widely used extensively tested geochemical code PHREEQC (Parkhurst and Appelo, 2013) to synthesize the experimental data to develop and parameterize mechanistic representations. The implementation of CLM-CN reactions in a geochemical code is detailed elsewhere (Tang et al., 2016). Guidelines for implementation of the microbial reactions, surface complexation, WHAM, etc., in PHREEQC are available in the user manual (Parkhurst and Appelo, 2013).

### 2.3.2 Parameterization

The stoichiometric and kinetic rate parameters for the CLM-CN reaction network are specified in Fig. 1. The indirect respiration faction $s_{labile}$ is highly uncertain. We start with $s_{labile} = 0.4$, and check the sensitivity with $s_{labile} = 0.2$ and 0.6. For the decay of biomass, and growth of methanogens, we use the general parameter values in the literature (Rittmann and McCarty, 2001). The half saturation $k_D$ and $k_{surf}$ values are taken from published literature (Jin and Roden, 2011). The parameter values and the references are listed in Table 1.

### 2.3.3 Initialization

The basic experimental parameters are summarized in Table 2 and Table S1. The amount of water, headspace volume, and temperature are set at the experimental parameter values. The initial pH, organic acids (combined formate, acetate, propionate, and butyrate from Table S1 to Table 2) and Fe(II) concentration are specified as measured.

The measured total organic carbon includes seven carbon pools in the CLM-CN decomposition cascade, as well as simple substrates (such as sugars, alcohols, organic acids), and biomass for FeRB, MeGA, MeGH, and other microbes. Because of the lack of reliable methods in partitioning the measured total organic carbon into these pools, we combine the Lit1 pool with LabileDOC, Lit2 with SOM1, and Lit3 with SOM2 pools as they have identical turnover times (Fig. 1). That is, we will split the initial total organic carbon (minus simple substrates) into LabileDOC, SOM1, SOM2, SOM3, SOM4, FeRB, MeGA, and MeGH pools, with fraction $f_{LabileDOC}$, $f_{SOM1}$, $f_{SOM2}$, $f_{SOM3}$, $f_{FeRB}$, $f_{MeGA}$, and $f_{MeGH}$ (the rest is $f_{SOM4}$, i.e. $f_{SOM4} = 1 - f_{LabileDOC} - f_{SOM1}$ - ...). Because the experiments lasted for only 2 months, and predictions are often not very sensitive to the initial biomass (Tang et al., 2013b; Tang et al., 2013c; Xu et al., 2015; Jin and Roden, 2011), the predictions are expected to

be sensitive to $f_{\text{LabileDOC}}$, $f_{\text{SOM1}}$, and $f_{\text{SOM2}}$ under the experimental conditions (as the turnover time for SOM3 and SOM4 are 2 and 27 y, respectively, Fig. 1). With a turnover (mean residence) time of 0.2-0.5, 6-9, and >125 years for the fast, slow, and passive pools, respectively, less than 5 % was estimated for the fast pool for 121 individual samples from 23 high-latitude ecosystems located across the northern circumpolar permafrost zone (Schädel et al., 2014). Based on incubation tests with Siberian soils for over 1200 d, the initial labile carbon pools were estimated to comprise $2.22 \pm 1.19$ and $0.64 \pm 0.28$ % of the total organic carbon with turnover times of $0.26\pm1.56$ and $0.21\pm1.58$ y under aerobic and anaerobic conditions, respectively (Knoblauch et al., 2013). We set $f_{\text{LabileDOC}} = 0.0005$, $f_{\text{SOM1}} = 0.01$, $f_{\text{SOM2}} = 0.02$, $f_{\text{SOM3}} = 0.1$, $f_{\text{bio}} = 10^{-6}$, $f_{\text{MeGA}} = f_{\text{MeGH}} = f_{\text{bio}}$, and $f_{\text{ferb}} = 2 f_{\text{bio}}$ [$f_{\text{bio}} = 10^{-6}$ approximating with *E. coli* with a wet weight $10^{-12}$ g, 70 % water, and 50 % dry weight carbon (Madigan, 2012), each microbial cell contains $\sim 1.25\times10^{-14}$ mole C, this means $\sim 10^8$ cells in 1 mole total organic carbon, which roughly approximates the range of reported values (Roy Chowdhury et al., 2015)].

Bioavailable ferric oxides are assumed to be in the form of $Fe(OH)_{3a}$, with initial concentration as a fraction $f_{\text{fe3}}$ of the dry soil mass. Depending on the season and the age of the drained thawed lake basins, HCl extractable Fe(III) is reported to range between 100 and 700 g Fe(III) m$^{-3}$ in the Barrow soils in a 24 cm soil profile (Lipson et al., 2013a). Using a weighted average of bulk density of 0.26, this translates to 0.2 to 1 % g Fe(III)/g dry soil mass. While bioavailable Fe(III) in soils is not well defined (e.g., Hyacinthe et al. 2006; Poulton and Canfield 2005), we start with $f_{\text{Fe3}} = 0.005$ and evaluate the impact with a range of values. Fe(III) reduction dissolves $Fe(OH)_{3a}$ and releases adsorbed protons on the mineral surface, which is described by the surface complexation model (Dzombak and Morel, 1990). The organic content for WHAM is set at total organic carbon. The initial total inorganic carbon (TIC) in the solution is assumed to be in equilibrium with an atmosphere of $CO_2$ at 400 ppm and 1 atm. The headspace gas starts with $N_2$ at 1 atm. These parameters are summarized in Table S2. Additional specifics are available in the scripts to produce input files. The reaction database [extended from (Tang et al., 2013b; Tang et al., 2013c)], the python scripts to create input files for various locations, temperatures, and other options (e.g., temperature and pH response functions) and scripts used to make the figures are provided as supplemental information.

## 3 Results and discussion

### 3.1 Experimental observations

The experimental results of anoxic soil incubation experiments were published elsewhere (Herndon et al., 2015a; Roy Chowdhury et al., 2015), so we briefly describe the original observed headspace $CO_2$ and $CH_4$ concentration, soil Fe(II) and organic acids concentration, and pH (Fig. 2). The variations in the overall observations appear to be better explained by the differences between the soil horizons (organic vs. mineral soils) than among the microtopographic locations (center, ridge, and trough) of ice-wedge polygons. Up to 20 % $CO_2$ was observed in the headspace by the end of the 2-month incubations, with higher concentrations in the organic soils than in the mineral soils (Fig. 2a1-3 vs. 4-6). This can be attributed to the higher organic content of the organic soils compared to that of the mineral soils (Table 2, Table S1).

$CO_2$ in the headspace increased rapidly in the beginning and then the increase slowed (Fig. 2). The initial rapid increase can be attributed to fast decomposition of the easily degradable substrates such as sugars, alcohols, etc. (Yang et al., 2016; Fey and Conrad, 2003; Glissmann and Conrad, 2002; Kotsyurbenko et al., 1993). As the easily degradable substrates were exhausted, the $CO_2$ production rate decreased. These observations are similar to those for the anaerobic incubations with soils from a trough location in a high center polygon at the same site (Yang et al., 2016) and deep Siberian permafrost soils (Knoblauch et al., 2013). However, $CO_2$ continued to increase well beyond 2 months in both these previous studies , and the $CO_2$ production rates stabilized, probably reaching a rate limited by the slow rate of hydrolysis in the Siberian soil microcosms. These observations are different from the observed $CO_2$ level off in the current microcosms (Fig. 2a2, a4, a5).

$CH_4$ in the headspace increased slowly at the beginning and then accelerated (Fig. 2b1-5), except the center organic soils. $CH_4$ accumulation lagged behind $CO_2$ for about 10 d in most of the microcosms and by a few days for the center organic soil microcosms at 4 and 8 °C. These lag times are shorter than those observed in microcosms with deep Siberian permafrost soils (average $960 \pm 300$ d) (Knoblauch et al., 2013). This is probably because of the initial abundance of substrates such as organic acids in the Barrow soils (Fig. 2c1-6). In addition, the shallow Barrow soils experience freezing and thawing, and so does microbial activity every year, while the deep Siberian permafrost soils were frozen for extended periods; as a result, the amount of initial biomass in the shallow Barrow soils is probably much higher than that in the deep Siberian soils.

Organic acids generally accumulated at the beginning, decreased as $CH_4$ concentration increased, and exhausted in the mineral soil microcosms (Fig. 2c1-6). In contrast, organic acids were not exhausted in the center organic soil microcosms (Fig. 2c6). In comparison with similar tests with soils from the high center polygon trough, organic acids accumulated for over 5 months in the organic soils and were not exhausted in the mineral soils (Yang et al., 2016). The accumulation and disappearance of organic acids have been widely observed in the literature (van Bodegom and Stams, 1999; Fey et al., 2004; Glissmann and Conrad, 2002; Jerman et al., 2009; Kotsyurbenko et al., 1993; Lu et al., 2015; Peters and Conrad, 1996; Yao and Conrad, 1999).

Fe(II) concentrations increased and levelled off (Fig. 2d1-6), with similar trends for pH (Fig. 2e1-6). The increase in pH concurred with Fe(III) reduction, which released hydroxides from $Fe(OH)_{3a}$ dissolution. The pH increase is in contrast to the observed pH decrease when Fe(III) reduction was absent (Xu et al., 2015). While Fe(III) reduction was reported to inhibit methanogenesis through direct inhibition (van Bodegom et al., 2004) or substrate competition (Miller et al., 2015; Reiche et al., 2008), the impact appears less significant than expected in these incubations, as well as incubations with the high center polygon trough soils (Yang et al., 2016). This is consistent with the observation that methane production initiated in the presence of oxidants (Roy et al., 1997). In addition, Fe(III) reduction can both inhibit and promote methanogenesis (Zhuang

et al., 2015). In the Barrow soils, the initial abundance of organic acids probably mitigates the competition between Fe(III) reducing and methanogenic populations, decreasing the lag time between $CH_4$ and $CO_2$ accumulation.

Substantial microbial activity was observed at -2 °C, which is above the soil water freezing point due to osmotic and matric potentials. These incubations led to an increase in $CO_2$ (Fig. 2a1-6), organic acids (Fig. 2c1-6), Fe(II) (Fig. 2d1-6), and pH (Fig. 2e1-6). $CH_4$ concentrations were low but detectable in the headspace at -2 °C. The lag time between $CH_4$ and $CO_2$ increases with decreasing temperature, which was widely observed in the literature as well (Fey and Conrad, 2003; Hoj et al., 2007; Jerman et al., 2009; van Bodegom and Scholten, 2001; Fey et al., 2004; Kotsyurbenko et al., 1993; Lu et al., 2015). The transition from -2 to 4 and 8 °C appears to be gradual except for the center organic soils, where $CH_4$ increases were drastic from -2 to 4 °C (Fig.2a1 vs. b1). The observed overall temperature responses are diverse, as manifested by $Q_{10}$ values from 1.6 to 22 (Roy Chowdhury et al., 2015).

### 3.2 Modeling results

#### 3.2.1 Overall

With the same model parameter values given in Table 1 and Table S2 and different experimental parameter values listed in Table 2, the model roughly predicts the observed trends for different soils at the three temperatures (Fig. 2): $CO_2$ and $CH_4$ accumulate in the headspace; $CO_2$ accumulation slows down while $CH_4$ speeds up at later times; $CH_4$ lags behind $CO_2$; organic acids accumulate and then decrease; Fe(II) accumulates and levels off; pH increases and levels off; and carbon mineralization and methanogenesis rates increase with temperature.

While the model predicts little $CO_2$ and $CH_4$ in the headspace at -2 °C, which is similar to what was observed, it predicts little change in Fe(II) and pH as well, which is not consistent with the observations. To improve the prediction at -2 °C, which can be important (Zona et al., 2016; Xu et al., 2016a), it is necessary to understand why little $CO_2$ or $CH_4$ was observed to occur with Fe(III) reduction, which was indicated by the increase of Fe(II) and pH.

The same model parameter values describe the observed differences in the mineral soils better than in the organic soils. For the mineral soils, the model overpredicts the increasing trend for $CO_2$ in the headspace at late times because the observations levelled off (Fig. 2a1-3). The initial rapid $CO_2$ increases lasted for over 2 months in the 3-year incubations with Siberian permafrost soils under 4 °C and anaerobic conditions (Knoblauch et al., 2013). In these long-term tests, $CO_2$ increased rapidly at the beginning and the rate stabilized as the carbon release became limited likely by hydrolysis of polymers. The observed sustained $CO_2$ accumulation in these closed microcosms indicates that the observed trends in Fig. 2a1-6 at later times are probably uncertain. Except for these mismatches, the model predictions generally agree with the observations for the mineral soils reasonably well.

In contrast, the predictions do not agree as well with the observations for the organic soils. For the trough organic soils, the model underpredicts $CO_2$ in the headspace (Fig. 2a4) but describes the rest of the observations reasonably well. In addition to $CO_2$ (Fig. 2a5), the model underpredicts Fe(II) and pH increase in the ridge organic soils (Fig 2d5, e5). The prediction of the center organic soils differs from the observations the most (last column in Fig. 2). These mismatches might be explained by model biases in initial Fe(III) content, LabileDOC, and biomasses.

### 3.2.2 Fe(III) reduction

Agreement between predictions and observations for the Fe(II) and pH increase can be improved for the ridge and center organic soils by increasing the Fe(III) content from $f_{Fe3} = 0.005$ to 0.01 and 0.02 (Fig. 2d5-6, e5-6). This also increases the predicted $CO_2$ and $CH_4$ for the center organic soils (Fig. 2a6, b6) because of the predicted pH increase (Fig. 2e6), which increases the reaction rates as the pH response function increases when the calculated pH increases toward an optimal pH of 6.2 in Eq. (3). For the ridge organic soils, $f_{Fe3} = 0.01$ increases the predicted $CH_4$ like the center organic soils, but $f_{Fe3} = 0.02$ decreases $CH_4$ prediction because of the competition between methanogens and iron reducers and limited availability of substrates (Fig. 2b5). This provides an explanation as to why Fe(III) reduction can both suppress and promote methanogenesis (rather than strict thermodynamic control, e.g., Bethke et al., 2011; direct inhibition, e.g., van Bodegom et al., 2004; or indirect inhibition through substrate competition, e.g., Mill et al., 2015, Reiche et al. 2008).

As the bioavailable Fe(III) in the organic soils is reported to range from 0.2 to 1 % of dry soil mass (Lipson et al., 2013a), the short-term tests are not expected to be Fe(III) limited for the mineral soils. Increasing bioavailable Fe(III) makes the model overpredict Fe(II) and pH increases at later times for the mineral soils (Fig. 2d1-5, e1-4), and Fe(III) reduction and methanogenesis at later times are predicted to be limited by organic substrate availability at 4 and 8 °C (Fig. 2b1-4). The latter is consistent with the observed very low organic acids concentrations at the end (Fig. 2c1-5). As a result, the model underpredicts $CH_4$ accumulation, indicating the current parameterizations, in particular the half-saturation and growth rate constants, may over-predict the ability of iron reducing bacteria to outcompete methanogens.

### 3.2.3 $CO_2$ distribution among gas, aqueous, and adsorbed phases

While increasing Fe(III) slightly increases the predicted $CO_2$ for ridge mineral soils (Fig. 2a2), it decreases the predicted $CO_2$ in the headspace for trough and center mineral soils (Fig. 2a1 and a3). This is because $CO_2$ solubility is predicted to increase significantly as pH increases, resulting in the dissolution of $CO_2$ from the headspace into the aqueous phase (Fig. S1). To examine this impact, we conduct numerical simulations with a 45 mL headspace with an initial 1 atm $N_2$ gas and 10 mL solution with 10 mM total inorganic carbon at various temperature and pH values. $CO_2(g)$ and $CO_2(aq)$ or carbonic acid dominate at a pH lower than 5 (Fig. 3). As the pH increases above the carbonic acid pKa (around 6.3 at standard condition),

$CO_2(g)$ in the headspace and $CO_{2(aq)}$ decrease as $HCO_3^-$ becomes dominant in the aqueous phase, and the gas-phase fraction decreases dramatically. The gas-phase fraction also decreases with decreasing temperatures (Fig. 3).

In addition, $CO_2$ was reported to adsorb to surface sites (Appelo et al., 2002; van Geen et al., 1994; Villalobos and Leckie, 2000). With the surface complexation reactions between $Fe(OH)_{3a}$ and carbonate species, we add 1 mmole $Fe(OH)_{3a}$ (about the mean values in Fig. 2 for the case $f_{Fe3} = 0.02$) to the abovementioned numerical experiments. The calculations show that the adsorption phase can dominate at low pH (Fig. S2), with the total amount dependent on the abundance of surface sites. For the high-temperature high-Fe(III) initial content cases in Fig. 2, adding $CO_2$ sorption reactions provides a substantial buffer against the early increase in $CO_2$ in the headspace (Fig. S3). As the $Fe(OH)_{3a}$ is reduced and dissolved, the adsorbed $CO_2$ is predicted to be released, contributing to an increase in headspace $CO_2$ increase later on.

In addition to pressure, these calculations suggest the need to appropriately account for pH and its impact on the gas, aqueous, and adsorbed phases $CO_2$ partition when we use headspace concentration measurements from anaerobic incubations to estimate $CO_2$ emission. Otherwise, substantial uncertainties can be introduced. A geochemical model with accurate thermodynamic data and accounting for $CO_2$ sorption can be useful in accurately quantifying $CO_2$ production in these closed microcosms.

### 3.2.4 Initial $CO_2$ accumulation in the organic soil microcosms

The model underpredicts the early $CO_2$ increase in the headspace for the organic soil microcosms (Fig. 2a4-6), which is mostly apparent in the center organic soil microcosms. The reason is that the organic soil microcosms contain more labile organic carbon than the mineral soil microcosms, as evidenced by water extractable organic carbon (Table 2). In particular, the center organic soil microcosms contain about half total organic carbon of the other microcosms, double the water volume, and three to five times water extractable carbon (Table 2). As a result, it produces the most $CO_2$ and $CH_4$ and has a very short lag time between $CH_4$ and $CO_2$. If we increase the initial LabileDOC content $f_{LabileDOC}$ from 0.0005, as shown in Fig. 2, to 0.01, and 0.02 for the organic soil microcosms, the underprediction of the early $CO_2$ increase in the headspace are more or less mitigated (Fig. 4).

The predicted rapid initial $CO_2$ increase is due to the fast fermentation reactions (Fig. S4a1-6, e1-6). The predicted steep transition in $CO_2$ concentration increases appears reasonable for the center and trough soil microcosms, but less so for the ridge soil microcosms. In addition to the 20 h and 14 d turnover time differences, fermentation reactions decrease the pH, further inhibit the predicted SOM1 decomposition reactions, Fe(III) reduction, and methanogenesis, making the predicted transition steeper. The fast fermentation is consistent with the observed rapid disappearance of glucose and increase of $CO_2$ after glucose addition in similar experiments with soils from a high center polygon trough from the same site (Yang et al., 2016). However, the observed decrease of natural free reducing sugars is gradual, with about one-third of the original

reducing sugars left over after 150 d of incubations. Along with the predicted rapid initial LabileDOC decrease and $CO_2$ increase, the model predicts a rapid initial increase in organic acids, which is close to the observations for the center soil microcosms but much greater than the observations for the trough and ridge soil microcosms. The latter indicates that the ratio of organic acids to $CO_2$ of 2:1 from the fermentation reaction (R1) may not be accurately representative of the experiments.

Detailed measurements showed a rapid initial increase and then a quick decrease of organic acids in the mineral soil microcosms and a gradual increase and slow decrease in the organic soil microcosms from a trough location in a high center polygon in the first 144 d anaerobic incubations (Yang et al., 2016). The rise and fall were fast in both the mineral and organic soil microcosms for ethanol, and were generally more gradual for organic acids than for ethanol (Yang et al., 2016). To explain the various observations for the organic soil microcosms and for accurate predictions, the diversity of the hydrolysis products (Feng and Simpson, 2008), and the subsequent pathways (Tveit et al., 2015) may need to be accounted for. Additional detailed data are needed to support increasingly mechanistic models, e.g., with reducing sugars to represent less rapid fermentation, and additional specific organic acids such as propionate and butyrate to better describe diverse observations in the incubations.

### 3.2.5 Carbon mineralization

Less than 1 % of the total initial carbon turned over to $CO_2$ and $CH_4$ in about 2 months, which is attributed mostly to decomposition of labile SOM (SOM1), LabileDOC, and organic acids (Fig. S4). Few changes are predicted in the slow pools (SOM3, and SOM4, not shown) even though they comprise a large portion of the soil carbon pool. The small amount of respired carbon is similar to the incubation tests conducted with Siberian permafrost soils under 4 °C, which was estimated to be 3.1 % and 0.55 % under aerobic and anaerobic conditions for 1200 d (Knoblauch et al., 2013), the 1-year aerobic incubations tests (Feng and Simpson, 2008), and the incubations from a wide range of Arctic soils (Schädel et al., 2014). All of these results suggest that the hydrolysis of macromolecular organics by extracellular enzymes could be a rate-limiting step at late times. To predict the long-term vulnerability of the organic carbons, it is important to understand and describe the hydrolysis of macromolecular components in SOM.

### 3.2.6 $CH_4$ accumulation

Besides Fe(III) reduction, the predicted $CH_4$ production is dependent on the substrate production. With $s_{labile} = 0.2$, the model generally predicts less $CH_4$ and more $CO_2$ than the case with $s_{labile} = 0.4$ because less SOM is assumed to respire through the anaerobic pathway in the $s_{labile} = 0.2$ case (Fig. S5). With increased $s_{labile} = 0.6$, the model predict more $CH_4$ and less $CO_2$. The impact on the mineral soils is generally more pronounced than the organic soils because the former is more substrate limiting than the latter. Unlike $CO_2$, $CH_4$ solubility and adsorption are much lower. Gas-phase $CH_4$ in the headspace dominates over aqueous and adsorbed phases. The model predicts the general exponential increase trend with a lag time behind $CO_2$ (Fig. 2).

However, the prediction is sensitive to Fe(III) reduction, pH, temperature (Fig. 2), and labile substrates (Fig. 4). The model substantially underpredicts early fast $CH_4$ production for the center organic soil microcosms (Fig. 4b3). While the cell count for the center organic soils is not available for day 0, the data did show that the center organic soils had the highest amount of biomass after 100 d incubations (Roy Chowdhury et al., 2015). The disagreement between the predictions and the observations can be mitigated by increasing the initial biomass $f_{bio}$ from $10^{-6}$ to $10^{-5}$ and $2 \times 10^{-5}$ for the center organic soil microcosms (Fig. 5). With increased initial biomass, Fe(III) reduction and methanogenesis are predicted to speed up the recovery of the initial pH drop caused by organic acids accumulation so that the model predicts a fast $CH_4$ increase that is comparable to the observed. However, the model overpredicts the $CH_4$ increase at late time, indicating alternative inhibition mechanisms rather than substrate limitation on methanogenesis at late time or additional $CH_4$ consumption such as anaerobic oxidation (Caldwell et al., 2008; Smemo and Yavitt, 2011).

### 3.2.7 pH

With the complexation reactions involving proton or hydroxide anion with carbonate species, ferrihydrite surface, and SOM, the geochemical model describes the observed pH evolution reasonably well (Fig. 2). The initial pH was lower in the mineral soils than in the organic soils (Fig. 2), probably because of less buffering capacity due to less organic matter in the mineral soils and/or more reducing condition in the organic soils as reduction reactions typically consume protons. Because the ridge mineral soils have the lowest initial pH, the CLM4Me pH factor is the lowest (Table S1), contributing to the underprediction of $CH_4$ (Fig. 2b2). With high organic content, the organic matter dominates the aqueous geochemistry, and the predicted pH is sensitive to the surface sites specified for WHAM. If the specified WHAM organic matter is reduced by 25 %, then the pH buffering capacity is decreased and the predicted pH increases substantially (Fig. S6e1-6) even though the predicted changes in organic acids and Fe(II) are small. For the trough soils, the predicted pH surpasses the optimal of 6.2, and f(pH) (Eq. 3) decreases (Fig. S6e1, e4). As a result, predicted $CO_2$ and $CH_4$ are decreased. The pH impact becomes complex around the optimal pH. If we increase the specified WHAM organic matter by 25 %, the predicted pH is lower due to larger pH buffering and the reaction rates are generally smaller. Setting the WHAM sites at measured total organic carbon works reasonably well for the experiments with the CLM4Me pH response function.

Comparing the CLM4Me pH response function with these used in TEM and DLEM, all three response functions show that the reaction rates are sensitive to pH (Fig. 6), which is expected to influence the predictions for these incubation tests as the pH increases from about 5.5 to 7. In this range, CLM4Me and DLEM have a similar slope, but the latter has a greater rate reduction effect. While CLM4Me and TEM have a similar rate reduction effect, CLM4Me has a steeper curve than TEM. These differences translate to substantial differences in model predictions (Fig. S7). All calculated f(pH) values increase during the tests (Fig. S7f1-f6). As the f(pH) calculated by DLEM is the lowest, the predicted changes are the smallest. The f(pH) calculated by TEM is slightly greater than CLM4Me at the beginning and is the opposite at late times (Fig. 6). As a result, TEM generally predicts slightly faster evolution than CLM4Me as the reaction rates at the late time are limited by

substrates rather than pH. While the pH ranges from 3.3 to 8.6 in the Arctic soils (Schädel et al., 2014), the range and the variability of the data are limited in the evaluation of these pH response functions. Nevertheless, model predictions are sensitive to pH response functions; the microbes are likely adapted to the site pH conditions such that the response functions are expected to vary among sites and functional groups. Therefore, pH response function can be an important source of prediction uncertainty.

### 3.2.8 Temperature response

Temperature effects on reactions between inorganic aqueous species, and the aqueous and gas species, are taken into account in the established reaction database. The temperature impact on surface complexation reactions with ferric hydrous oxides, and with SOM in WHAM is not quantified, which can be a potential source of uncertainty. LSMs generally use empirical (e.g., CLM-CN, CENTURY), $Q_{10}$, or the Arrhenius equations. The CLM-CN temperature response function is compared with the CENTURY, $Q_{10}$ equation, Arrhenius equation, and Ratkowsky equation in Fig. 7 and Fig. S8. All of these temperature response functions describe increasing rate with increasing temperature. When the temperature response functions $f$(T) are plotted in arithmetical scale, the shapes are similar except for CENTURY, which approaches 1 when the temperature increases above 20 °C; CLM-CN is close to $Q_{10}$ with $Q_{10}$ = 2.5, the Arrhenius equation with $E_a$ = 60 kJ mol$^{-1}$ and the Ratkowsky equation with $T_m$ = 260 K. When f($T$) is plotted in log scale (Fig. 7), $Q_{10}$ and Arrhenius equations are approximately linear while the rest have a similar shape; CLM-CN appears close to Ratkowsky equation with $T_m$ = 260 K. At our temperatures -2, 4, and 8 °C, CLM-CN is very close to CENTURY, $Q_{10}$ = 2.5, $E_a$ = 60 kJ mol$^{-1}$, and $T_m$ = 260 K (Fig. 7, Fig. S8). Despite their closeness, the predictions can be different for the different response functions (Fig. S9, Fig. S10), reflecting the sensitivity of the temperature effect on the biogeochemical reaction rates. The difference is amplified when different $Q_{10}$, $E_a$, or $T_m$ is used (not shown), introducing potentially large uncertainty in model predictions. Because the temperature response functions are expected to vary for different micro-organisms, extra-cellular vs. intra-cellular enzymes, and geochemical reactions in the soil environment, improved quantification is needed.

### 3.2.9 Predicted impact of headspace gas accumulation

The accumulation of gases in the headspace may impact the soil carbon mineralization and methanogenesis. Knoblauch et al. (2013) and Yang et al. (2016) flushed the headspace of the microcosms while Roy Chowdhury et al. (2015) and Herndon et al. (2015) did not. The field conditions are likely somewhere between an open system and a closed system because neither the atmospheric pressure nor the hydrostatic pressure is constant, and the produced $CO_2$ and $CH_4$ are not always free to release to the atmosphere. To assess the impact of $CO_2$ accumulation in the headspace on the soil carbon mineralization and methanogenesis, we conduct numerical experiments with 10 and 100 times the headspace volume of the experimental values. With increased headspace volume, the headspace and aqueous $CO_2$ concentrations are predicted to decrease (Fig. S11 f1-6, g1-6), and the pH increase is predicted to slow down. As a result, the biogeochemical reaction rates are generally slower (Fig. S11e1-6). Eventually, the predicted total $CO_2$ and $CH_4$ production generally decrease with lower headspace $CO_2$

concentration (Fig. S11a1-6,b1-6). However, the impact on $CO_2$ production is very small for the organic soils in the trough and ridge location, and the $CO_2$ production is predicted to increase with decrease in headspace $CO_2$ concentration for the organic center soils. Because of the complicated nonlinear relationships in the biogeochemical processes, the impact of headspace gas accumulation on carbon mineralization and methanogenesis is not linear. While it is debatable that which

experimental conditions (flush the headspace or not) reflect the field conditions, biogeochemical models like ours provide a mechanistic method to account for this impact by using boundary conditions that reflect the reality. Additional targeted experiments and mechanistic models are necessary to better understand the impact under different conditions, and develop representations that reflect field conditions.

## 4 Summary and conclusion

Soil organic carbon turnover and $CO_2$ and $CH_4$ production are sensitive to redox potential and pH. However, land surface models typically do not explicitly simulate the redox or pH, particularly in the aqueous phase, introducing uncertainty in greenhouse gas predictions. To account for the impact of availability of electron acceptors other than $O_2$ on soil organic matter (SOM) decomposition and methanogenesis, we extend the CLM-CN decomposition cascade to link complex polymers with simple substrates and add Fe(III) reduction and methanogenesis reactions. Because pH was observed to

change substantially in the laboratory incubation tests and in the field and is a sensitive environmental variable for biogeochemical processes, we use the Windermere Humic Aqueous Model (WHAM) to simulate pH buffering by SOM. To account for the speciation of $CO_2$ among gas, aqueous, and solid (adsorbed) phases under varying pH, temperature, and pressure values, and the impact on typically measured headspace concentration, we use a geochemical model and an established reaction database to describe observations in recent anaerobic microcosms. Our results demonstrate the efficacy

of using geochemical models to mechanistically represent the soil biogeochemical processes for Earth system models.

Together with the speciation reactions from the established geochemical database and surface complexation reactions for ferric hydrous oxides, WHAM enables us to approximately buffer an initial pH drop due to organic acid accumulation caused by fermentation and then a pH increase due to Fe(III) reduction and methanogenesis. The single input parameter for

WHAM is total organic carbon content, which is available in any SOM decomposition model. Therefore, adding WHAM does not necessitate any additional characterization. However, the temperature effects on surface complexation reactions with ferric hydrous oxides and organic matter may need to be further quantified.

The equilibrium geochemical speciation reactions predict a substantial increase in $CO_2$ solubility as the pH increases above

30 6.3 because the aqueous dominant species shifts from $CO_2$ to $HCO_3^-$. Adding $CO_2$ adsorption to surface sites of metal oxides further increases predicted solubility at low pH. Without taking into consideration of speciation, and pH, temperature and

pressure impact with a geochemical model, the carbon mineralization rate can be substantially underestimated from anaerobic microcosms based on headspace $CO_2$ measurements.

Because various microbes respond to the temperature and pH change differently, it is challenging to describe observed diverse responses with any single one of the existing response functions. As the microbes adapt to the low temperature and pH conditions in the Arctic, the optimal growth temperature and pH value in these response functions may need to be adjusted to account for biological acclimation.

We demonstrate that a geochemical model can mechanistically predict pH evolution and accounts for the impact of pH on biogeochemical reactions, which enhances our understanding of and ability to quantify the experimental observations. Because pH is an important environmental variable in the ecosystems and land surface models either specify a fixed pH or use simple empirical equations, a geochemical model has the potential to improve model predictability for greenhouse emissions by mechanistically representing the soil biogeochemical processes.

Another following up task can be assessing this new framework of anaerobic SOM decomposition in field studies with CLM-PFLOTRAN. This can be done incrementally, i.e., add/remove reactions one at a time without source code modifications. CLM-PFLOTRAN currently uses CLM4.5 vertical resolved grid. The resolution can be adjusted, possibly in three dimensions, to reflect the heterogeneity of any structural soil column to account for the limitation of electron donors and electron acceptors at individual locations. As we gradually implement more and more processes, such as gas and aqueous transport through soils and aerenchyma, explicitly representing microbial processes for carbon decomposition, we hope the new framework will be useful for future investigation and model developments.

**Code availability**

PHREEQC is publically available at http://wwwbrr.cr.usgs.gov/projects/GWC_coupled/phreeqc/.

**Data availability**

The experimental data, scripts to produce the PHREEQC input files, and plot the figures are archived at https://github.com/t6g/bgcs .

## Appendix A Additional pH response functions

With $pH_{min}$, $pH_{opt}$, and $pH_{max}$ of 4, 7 and 10 with no microbial activity at pH below $pH_{min}$ or above $pH_{max}$, the pH response function used in DLEM is (Tian et al., 2010)

$$f(pH) = \frac{1.02}{1.02 + 10^6 \exp{(-2.5pH)}},$$ (A1)

for pH < 7; otherwise,

$$f(pH) = \frac{1.02}{1.02 + 10^6 \exp{(-2.5(14 - pH))}}.$$ (A2)

TEM uses a bell-shaped function (Cao et al., 1995;Xu et al., 2015;Raich et al., 1991)

$$f(pH) = \frac{(pH - pH_{min})(pH - pH_{max})}{(pH - pH_{min})(pH - pH_{max}) - (pH - pH_{opt})pH_{opt}},$$ (A3)

with $pH_{min}$, $pH_{opt}$, and $pH_{max}$ = 5.5, 7.5, and 9, respectively (Cao et al., 1995). Considering the typical acidic conditions in the Arctic and wetlands, we use the DLEM parameter values (Tian et al., 2010) as substantial $CH_4$ was observed in the incubation tests below pH 5.5 (Roy Chowdhury et al., 2015).

## Appendix 2 Additional temperature response functions

The $Q_{10}$ method is the most common temperature response function used in LSMs (Xu et al., 2016b;Berrittella and Van Huissteden, 2009, 2011;Walter and Heimann, 2000;Zhuang et al., 2004;Riley et al., 2011;Oleson et al., 2013). It is

$$f(T) = Q_{10}^{\frac{T - T_{ref}}{10}},$$ (B1)

with $T_{ref}$ as a reference temperature usually at 25 °C. However, the $Q_{10}$ value varies from 1.5 to 28 (Segers, 1998;Mikan et al., 2002), which indicates inadequate representation of the supply of substrates (Davidson and Janssens, 2006;Davidson et al., 2006), and microbial functional groups (Blake et al., 2015;Svensson, 1984;Rivkina et al., 2007;Lu et al., 2015) and necessitates alternative temperature response functions.

The Arrhenius equation (Arah and Stephen, 1998;Wang et al., 2012;Grant, 1998;Grant et al., 1993;Sharpe and DeMichele, 1977;Grant and Roulet, 2002) is

$$f(T) = \exp\left[-\frac{E_a}{R}\left(\frac{1}{T} - \frac{1}{T_{ref}}\right)\right],$$ (B2)

with $E_a$ as the activation energy, and $R$ as the gas constant. It is related to the $Q_{10}$ method with $\ln(Q_{10}) = \frac{10E_a}{RT_{ref}T}$. The introduced variability by the absolute temperature $T$ is not able to explain the wide range of $Q_{10}$ values either. Consequently, empirical equations are often used (Nicolardot et al., 1994). DayCent, ForCent, and CENTURY use (Parton et al., 2010)

$$f(T) = 0.56 + 0.465 atan[0.097(T - 15.7)].$$ (B3)

A temperature response function for microbial growth is (Ratkowsky et al., 1982)

$$f(T) = \left(\frac{T - T_m}{T_{ref} - T_m}\right)^2,$$ (B4)

with $T_m$ as a conceptual temperature of no metabolic significance between 248-296 °K, depending on the bacterial cultures.

**Acknowledgements**

This research was funded by the U.S. Department of Energy, Office of Sciences, Biological and Environmental Research, Terrestrial Ecosystem Sciences Program, and is a product of the Next-Generation Ecosystem Experiments in the Arctic (NGEE-Arctic) project. ORNL is managed by UT-Battelle, LLC, for the U.S. Department of Energy under contract DE-AC05-00OR22725. XX is grateful for the supports from the San Diego State University.

**Disclaimer**

This manuscript has been authored by UT-Battelle, LLC under contract no. DE-AC05-00OR22725 with the U.S. Department of Energy. The United States Government retains and the publisher, by accepting the article for publication, acknowledges that the United States Government retains a non-exclusive, paid-up, irrevocable, worldwide license to publish or reproduce the published form of this manuscript, or allow others to do so, for United States Government purposes. The Department of Energy will provide public access to these results of federally sponsored research in accordance with the DOE Public Access Plan (http://energy.gov/downloads/doe-public-access-plan).

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

Table 1. Model parameter values for base scenario

| Reaction | $k_{max}$ (d$^{-1}$) | $k_D$ (μM) | $k_{surf}$ | Reported $k_{max}$ range |
|---|---|---|---|---|
| R1 | 0.83 | | | |
| R2 | 0.5 | 12[1] | 0.062[1] | 0.96-2.16[2], 0.55 and 2.38[3], 0.34[4] |
| R3 | 0.8 | 11[1] | 0.062[1] | |
| R4 | 0.3[5] | 23[1] | | |
| R5 | 0.5[5] | 4.7[1] | | |
| R6 | 0.05[5] | | | |

[1] Jin, Q., and Roden, E. E.: Microbial physiology-based model of ethanol metabolism in subsurface sediments, J. Contam. Hydrol., 125, 1-12, 10.1016/j.jconhyd.2011.04.002, 2011.

[2] Esteve-Núñez, A., Rothermich, M., Sharma, M., and Lovley, D.: Growth of Geobacter sulfurreducens under nutrient-limiting conditions in continuous culture, Environmental Microbiology, 7, 641-648, 10.1111/j.1462-2920.2005.00731.x, 2005.

[3] Cord-Ruwisch, R., Lovley, D. R., and Schink, B.: Growth of Geobacter sulfurreducens with Acetate in Syntrophic Cooperation with Hydrogen-Oxidizing Anaerobic Partners, Applied and Environmental Microbiology, 64, 2232-2236, 1998.

[4] Holmes, D. E., Giloteaux, L., Barlett, M., Chavan, M. A., Smith, J. A., Williams, K. H., Wilkins, M., Long, P., and Lovley, D. R.: Molecular Analysis of the In Situ Growth Rates of Subsurface Geobacter Species, ibid., 79, 1646-1653, 2013.

[5] Rittmann, B. E., and McCarty, P. L.: Environmental biotechnology: principles and applications, McGraw-Hill, 2001.

Table 2. Experimental parameter values summarized from (Herndon et al., 2015;Roy Chowdhury et al., 2015). TOTC = total organic carbon. WEOC = water extractable organic carbon.

| Location | Horizon | Depth (cm) | pH | Soil (dwt g) | Water (g) | TOTC (g) | WEOC (mg) | Organic acids (mgC) | Fe(II) (mmol) | Bulk den. (g/cm$^3$) | Headspace (ml) |
|----------|---------|------------|------|--------------|-----------|----------|-----------|---------------------|---------------|----------------------|----------------|
| Center | Oa | 6-21.5 | 5.02 | 1.412 | 13.588 | 0.542 | 9.585 | 2.079 | 0.0107 | 0.9106 | 42.5282 |
|  | Bgh | 21.5-53.5 | 4.84 | 9.146 | 5.854 | 1.260 | 3.845 | 0.394 | 0.1302 |  |  |
| Ridge | Oe | 0-8 | 5.21 | 3.212 | 11.788 | 1.249 | 6.790 | 0.016 | 0.0190 | 1.0003 | 44.0051 |
|  | Bh | 8-42 | 4.54 | 8.621 | 6.379 | 1.263 | 3.282 | 0.409 | 0.1466 |  |  |
| Trough | Oe | 0-19 | 5.23 | 4.310 | 10.690 | 0.886 | 3.324 | 0.022 | 0.1675 | 0.9724 | 43.5745 |
|  | Bh/ice | 25-69 | 4.95 | 8.380 | 6.620 | 0.670 | 2.013 | 0.292 | 0.0475 |  |  |

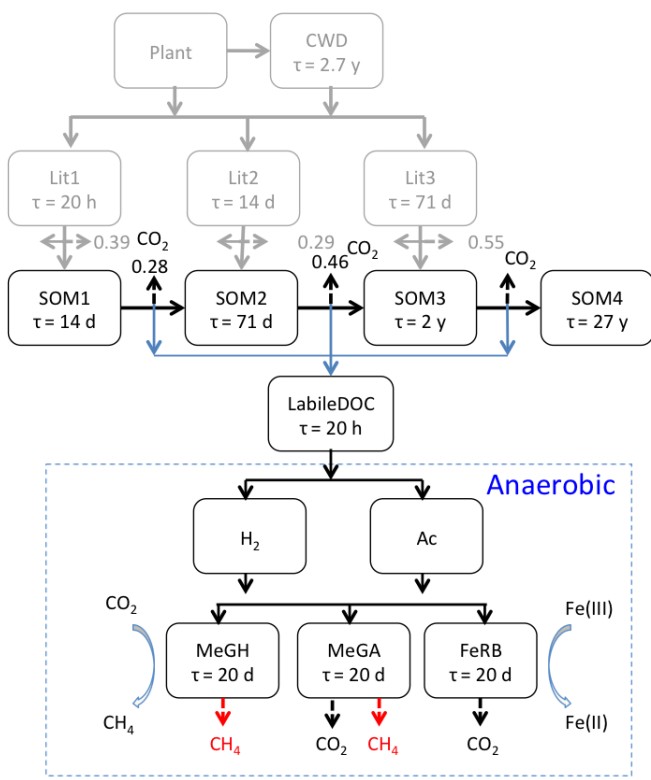

**Figure 1: Extension of the CLM-CN decomposition cascade (Thornton and Rosenbloom, 2005) to include a LabileDOC pool. A portion of the original respiration fraction is assumed to produce LabileDOC, which undergoes fermentation, Fe reduction and methanogenesis to release $CO_2$ and $CH_4$. FeRB, MeGA, and MeGH denote microbial mass pools for Fe reducers, acetoclastic and hydrogenotrophic methanogens, respectively. τ is the turnover time.**

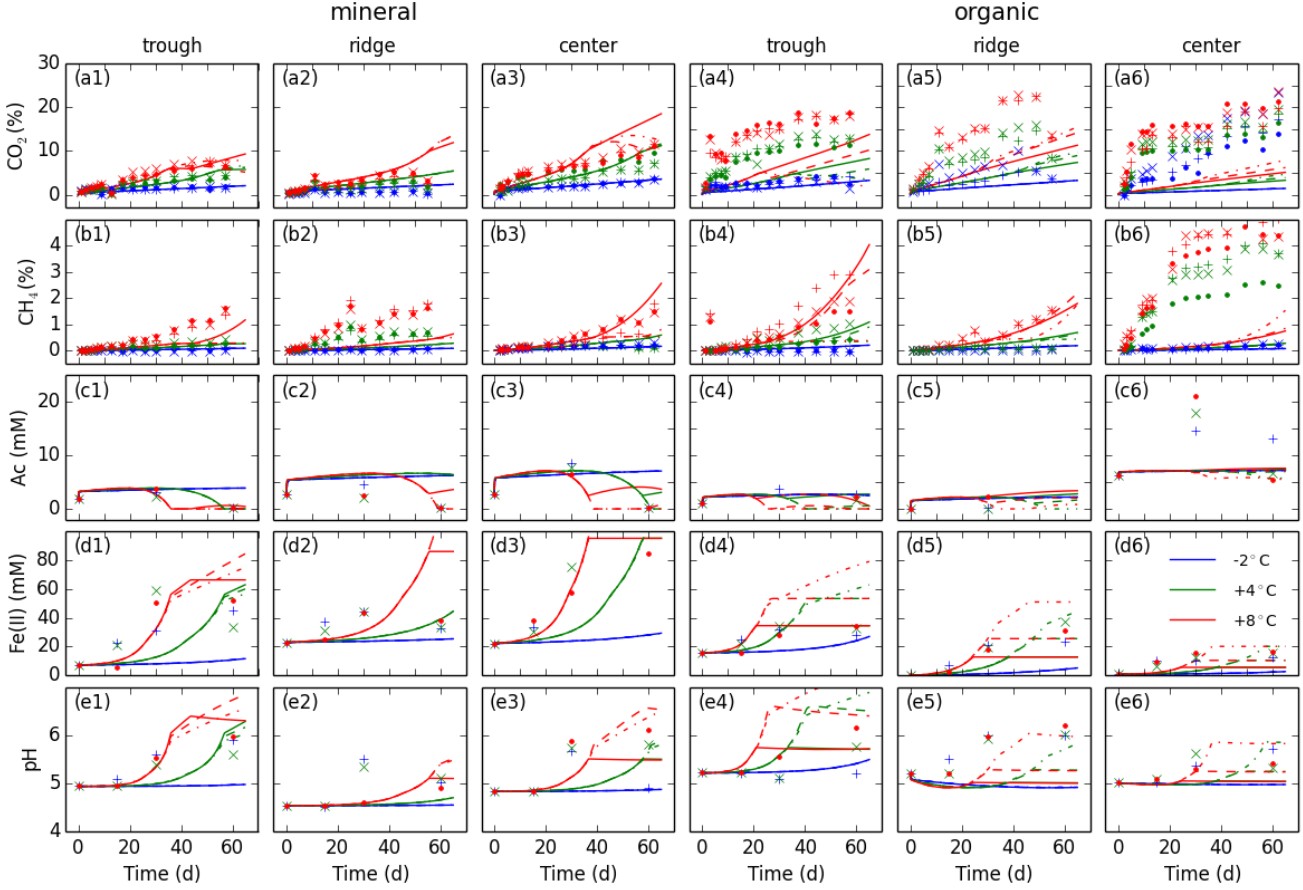

**Figure 2: Comparison of observed and modeled CO$_2$ (a1-6) and CH$_4$ (b1-6) in the headspace, organic acid (Ac, c1-6), extractable Fe(II) (d1-6), and pH (e1-6) in the incubation tests with soils from an Arctic lower center polygon. Symbols represent observations with blue, green and red for -2, 4, and 8 °C. For CO$_2$ and CH$_4$, different symbols of the same color represent duplicates. The organic acids, such as formate, acetate, propionate, and butyrate, reported by (Herndon et al., 2015) are combined as Ac in c1-6. The rest of the data were taken from (Roy Chowdhury et al., 2015). The curves are calculations based on model parameter values listed in Table 1 and experimental parameter values listed in Table 2. Trough, ridge, and center denote the microtopographic locations in the polygon, and mineral and organic denote soil horizons. Increasing the initial bioavailable Fe(III) $f_{Fe}$ from 0.005 (continuous) to 0.01 (dash) and 0.02 (dashdot) brings the predictions close to the observations for Fe(II) and pH for center and ridge organic soils.**

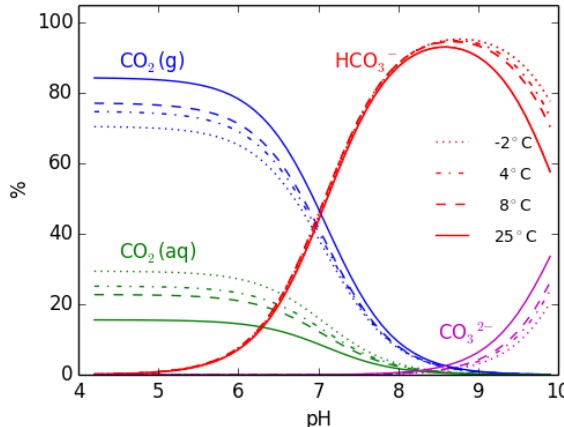

**Figure 3: Partition of CO$_2$ among gas and aqueous phase species under various temperatures. The calculations are conducted with 45 ml headspace with N$_2$ and 10 ml solution with 10 mM total inorganic carbon using PHREEQC. Gas phase dominates at lower pH and high temperature. As pH increases, gas phase CO$_2$ fraction is very low after pH 7, implicating potential underestimation of carbon mineralization based on headspace CO$_2$ concentration measurement only.**

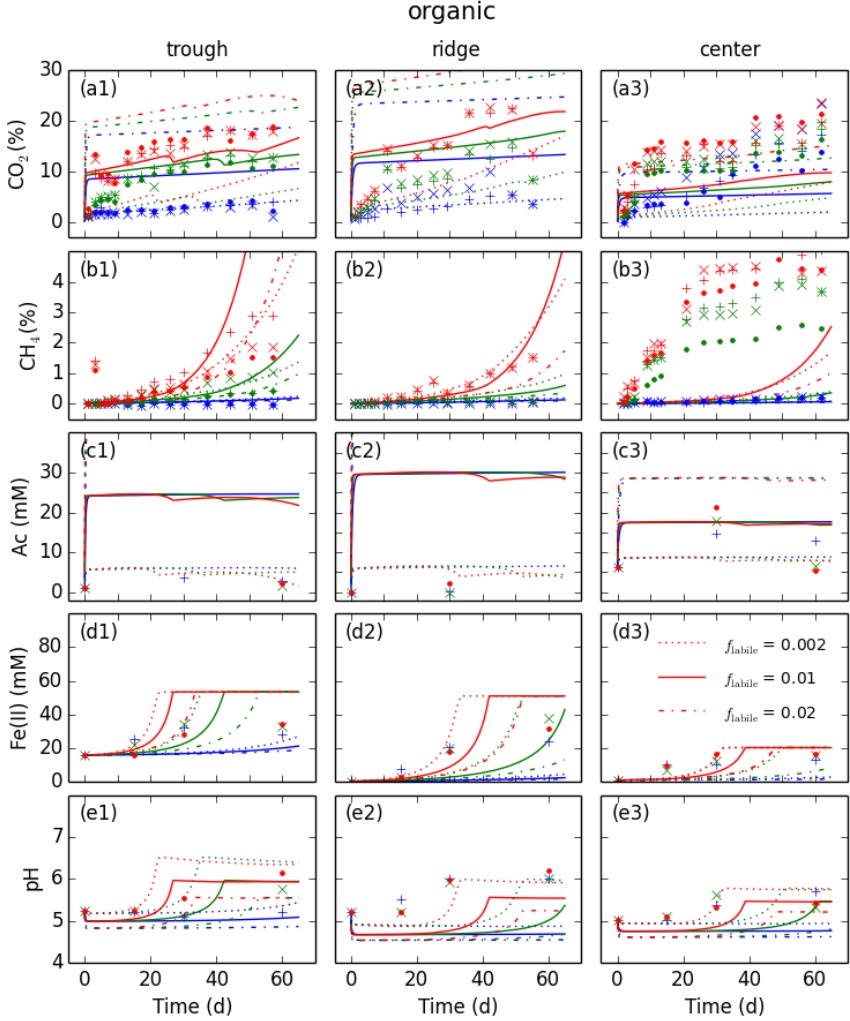

**Figure 4: Increasing initial LabileDOC better describes the observed initial rapid CO₂ increase in the headspace for the organic soils. See Figure 2 caption for more information.**

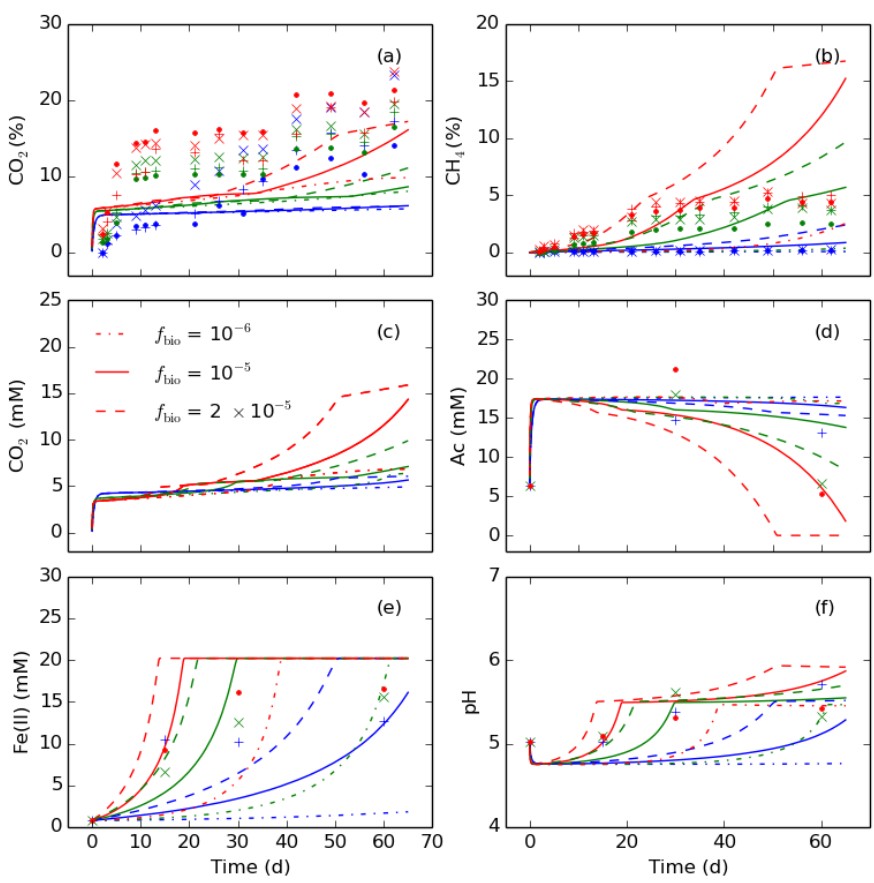

**Figure 5: Increasing the initial biomass predicts rapid CH$_4$ accumulation at early times that is close to the observations but misses the level-off trend at late time for the center organic soils. See Figure 2 caption for more information.**

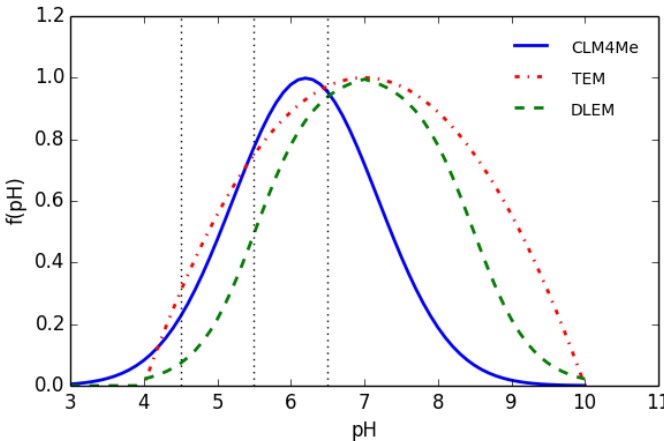

**Figure 6: Comparison of pH response functions used in CLM4Me (Riley et al., 2011), TEM (Raich et al., 1991), and DLEM (Tian et al., 2010) as described by Eqs. 3, A1-3. Reaction rates are sensitive to pH and pH response functions vary substantially, introducing prediction uncertainty.**

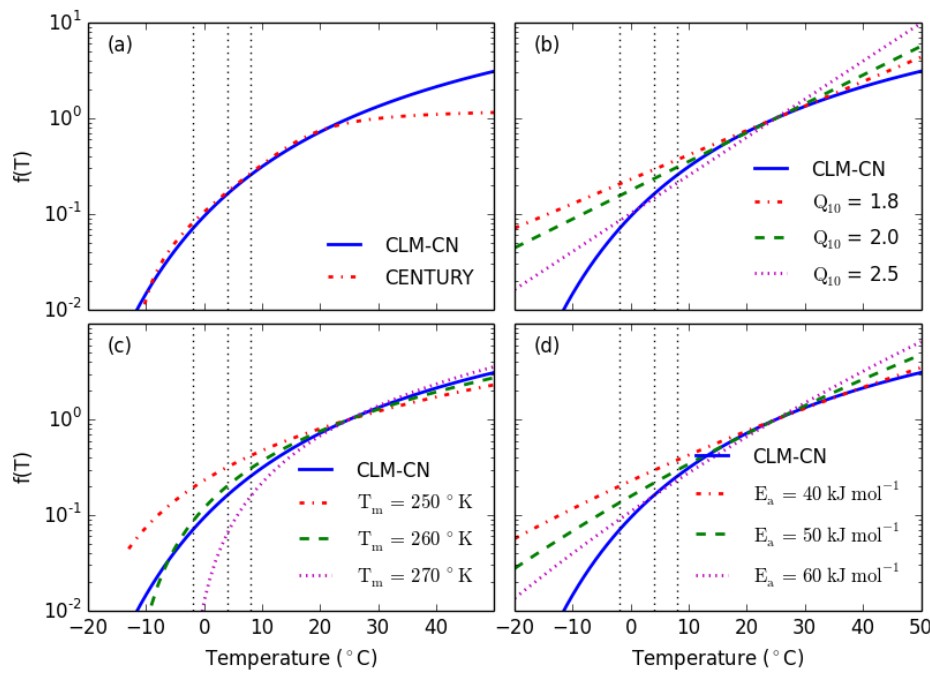

**Figure 7: Comparison of temperature response functions used in (a) land surface models CLM-CN (Thornton and Rosenbloom, 2005), CENTURY (Parton et al., 2010), (b) $Q_{10}$ (Oleson et al., 2013), (c) Ratkowsky equation (Ratkowsky et al., 1982) and (d) Arrhenius equation (Wang et al., 2013) described by Eq. (4, B1-B4). Reaction rates are sensitive to temperature and temperature response functions vary substantially, introducing prediction uncertainty.**

