# Peer review of "Biogeochemical modeling of CO2 and CH4 production in anoxic Arctic soil microcosms"

_Biogeosciences, 2016_

## Referee Comment (RC1) · Anonymous Referee #1 · 17 Jun 2016

Soil carbon models are a critical source of uncertainty in Earth system models both due to limitation in model-data and process representation. This manuscript addresses modeling hurdles in a key process (anaerobic decomposition) missing in most soil carbon models. This work is a timely, novel, and carefully conducted. However I am uncomfortable with the current introduction, justification, and implication presented in the manuscript. The actual results section is very strong but would suggest extensive revisions to the introduction and conclusion.

In general, I would suggest better connections between and within the introductory paragraphs. The main paragraphs jump around and paragraphs them selves lack coherent structure.

I would also like to see a discussion/acknowledgement of other mechanisms that could

influence anaerobic decomposition which were not captured by the proposed model. While not ALL processes can be included in a model, some acknowledgement of those missing processes and either how then can be incorporated into future work or how they influence current simulations are appropriate and highlight the limitations of the proposed model.

Relatedly the authors need to address how these detailed pool based kinetic models scale to well-structured heterogeneous soils. Great detail is gone into on the chemical processes governing methanogensis in the introduction but there is no discussion of how the physical structure of the soil plays into substrate and oxygen availability and the inherent limitations to applying mechanistic pool models to highly structured soil columns. While this is a common shortcoming of soil carbon models I feel that, given the level of processes detailed covered in the model, this is critical to address and justify utilization of a pool model with such explicit process representation. Minimum the authors need to acknowledge that the scaling of known kinetics from well-mixed experiments to highly structure soil cores is a relevant open question in the field.

The great strength of this manuscript is the highly detailed and careful analysis of the proposed model, grounded against a data set. This was very well done and I feel should provide the backbone of a new discussion section which could be extended to suggest potential follow up experiments based on the model results. However there is no formal model-data integration nor a comparison with an adequate number of data sets to justify this as a mature component of a new CLM module, as is implied by the current introduction and discussion. If this is the manuscript that the authors want to write then I would suggest more data sets, a formal data-model integration, and demonstration of improvement to previous ratio-based models. But the current analysis would be completely appropriate in a different context which focused more on implications to future experimental designs and long term model projections. I strongly encourage the authors to carefully consider an alternative framing of this very interesting study.

——Line by line response——

P2L2 Actually many of the IPCC models suggest that SOC will increase in the northern latitudes due to increases in inputs (Todd-Brown et al 2014), I would suggest softening this statement to reflect the huge uncertainty in the current state of the science. Less controversial would be a statement referring to a general ramping up of the entire carbon cycle in response to climate change, increases are expected in both primary production and decomposition. Whether the net effect will be to convert SOC to CO2/CH4 is highly debated.

P2L2-14 I like the content of this paragraph but it needs to be re-organized. There are three distinct topics in this paragraph which would be better served breaking them up/integrating with later parts of the manuscript: a review of expected high latitude SOC vulnerability to climate change, summary of CLM-CN representation of anaerobic conditions (coupling this with a general review of aerobic decomposition would not go amiss here but that is a soft suggestion), and the comparison to lab incubations.

P2L15 CH4 is critical not just because of it's high global warming potential but also because of it's emissions rate and residence time in the atmosphere. Please add some citations to reflect this.

P2L19 Why is this lag critical?

P2L25 Why is the CH4:CO2 ratio important? Maybe lead with this being a critical parameter for current models and then show how this is a dynamic response to the competing Redox ladder. I think this is where the authors are trying to go with this but it is lost in the paragraph. Could the proposed model be compared with the standard ratio model?

P2L33 Why is an aqueous phase essential for these calculations? Soil decomposition models are implicit descriptions of carbon dynamics anyway, why do we need an explicit representation of this process? Can a micro scale process like an explicit terminal electron acceptor model be simulated on the macro scale? I would suggest placing this study in the context of the increasing number of 'explicit' soil carbon dynamics models

(ex: Wieder et al 2015). These models may or may not increase the overall accuracy of predictive dynamics over a well-tuned traditional model but they can provide critical scientific insights into the process of soil decomposition. This introduction lacks this critical nuance and oversells the capabilities of the proposed model.

Sect2.2.1(and elsewhere) Please refer to model pools and other variables by name (variable) consistently throughout the manuscript, ex: organic acid pool (Ac). This reduces the need to refer back to tables/sections. Manuscripts are rarely read linearly and having to search for abbreviation definitions slows down reading.

P8L20 I applaud the authors for making their scripts available in the supplemental. Thank you.

P12 Nice job walking through potential drivers of model-data mismatch. These provide a rich pool of candidates for future investigations. I feel that this should be the main focus of the conclusions. Given that a single data set was used and no formal model-data integration done this model is not quite ready for a full land carbon model integration as is implied by the authors in the introduction and conclusions. What IS done quite elegantly is an analysis of several representations of potential mechanisms and how they influence overall carbon mineralization in the context of a common model structure.

P12 WEOC, TOTC are an unusual acronym in the field. Consider writing out the full name instead, I found myself forgetting what it stood for around here and having to go look it up. See previous comment about variable/pool references.

P13L19 This is a highly controversial statement that does not belong in the results section. While it is appropriate to highlight the relatively low amount of mineralized carbon there are several possible mechanisms for this that are unrelated to the chemical structure as suggested by this statement. Just because that is the explanation that fits into the model that is presented, it is not the only explanation (I'm thinking of various physical mechanisms like co-location and aggregate formation, as well as the

substrate rarity argument ala Allson 2006). Please move to the conclusion and soften this statement considerably.

P14L17 Cite the equation reference for f(pH)

P14L26 Given the noise generally inherent in these measurements I would hesitate to call this a 'substantial' difference. Could you can provide error bars for the data or some kind of significance testing.

Figures: In general, would it be possible to add error/uncertainty bars to the data points in the figures? This would place the modeled sensitivity in the context of the measurement error.

P16L5-7 WHAM is an aqueous pool model, claiming that there is no needed modifications when applying it to a well structured soil column seems a bit of a stretch.

Table 2 Formatting needs to be fixed for the table entries and I would suggest replacing TOTC with Total Organic Carbon and WEOC with Water Extractable Organic Carbon. OC is a common enough abbr. that it could be used here without explanation but TOTC and WEOC are not.

Figure and Table captions: Figure and table captions need to be able to stand alone in the manuscript, people will often scan the figures to get a sense of the results of the manuscript. Please expand the figure captions to more fully reflect the conclusions being illustrated here, this is particularly needed for the supplemental figures.

=====================

Todd-Brown, K. E. O., Randerson, J. T., Hopkins, F., Arora, V., Hajima, T., Jones, C., Shevliakova, E., Tjiputra, J., Volodin, E., Wu, T., Zhang, Q. and Allison, S. D.: Changes in soil organic carbon storage predicted by Earth system models during the 21st century, Biogeosciences, 11(8), 2341–2356, doi:10.5194/bg-11-2341-2014, 2014.

Wieder, W. R., Allison, S. D., Davidson, E. A., Georgiou, K., Hararuk, O., He, Y., Hopkins, F., Luo, Y., Smith, M. J., Sulman, B., Todd-Brown, K., Wang, Y.-P., Xia, J. and Xu, X.: Explicitly representing soil microbial processes in Earth system models, Global Biogeochem. Cycles, n/a–n/a, doi:10.1002/2015GB005188, 2015.

Steven D. Allison and Allison, S. D.: Brown Ground: A soil carbon analogue for the green world hypothesis?, Am. Nat., 167(5), 619–627, doi:10.1086/503443, 2006.

———————————————————

---

## Referee Comment (RC2) · Anonymous Referee #2 · 8 Jul 2016

The manuscript proposes a new module to the CLM-CN. It attempts to address a key limitation of existing land surface models in that the anaerobic decomposition is poorly represented. This work is novel and quite thorough. Most of my comments are technical except that I am not 100% comfortable with the testing dataset. It appears that the incubation experiment did not flush the headspace of microcosms. (This manuscript and the original publication both neglected to discuss flushing). It is well known that the accumulation of CO2 and other trace gases in headspace distorts gas diffusion and greatly impacts decomposition. The suitability of the testing dataset is worth a thorough discussion.

I am very intrigued by this model, especially its addition of geochemical components. From an empirical perspective, I agree that redox dynamics and the turnover of pH and single substrates are all key in understanding anoxic decomposition. However, it is still

unclear whether adding these processes make sense in modeling decomposition and trace gases production. Personally, I would like to see a comparison between the new model and CLM4ME/CLM-CN and evaluate whether the new model offers meaningful improvement. Other data sources could also be used to calibrate the model. These and other future directions should be discussed in details in the conclusion.

Specific or technical comments Title, would it be better to say biogeochemical _modeling_ or _A_ biogeochemical model of CO2 and CH4 production?

When citing references, please add a space after each semicolon.

P3L2, there is a disconnect between the topic sentence and the following text. The topic sentence introduces 'simple substrates' and their importance in modeling CH4, while the following sentences switched the focus away from simple substrates. This paragraph can be reorganized to improve the flow of thought.

L8, lignin should not be classified as polysaccharides.

L14, acetate and H2 have been added _to models_?

When introducing soil pH and its role in LSMs, it is worth noting that soil pH changes rapidly as a result of redox reactions. For instance, iron(III) reduction consumes protons and usually increases soil pH.

L25, use pH buffering capacity instead of pH buffer capacity. It would also be useful to define 'pH response functions'.

L26, logarithmic

P4L4, the last sentence is rather weak and does not connect with the next paragraph. A better transition is needed to highlight why it is necessary to compare temperature response functions.

In 2.2.2, would it be possible to discuss how iron reduction and methanogenesis interact in the model? I can see that both of them reply on mD and kD, but what determines

the partitioning of electron donors between these two processes? These details would be valuable to interpret the results later on (P9L28-35).

P6L26, Riley et al. (2011) in fact cited Meng et al. (2012) Biogeosciences, 9, 2793–2819 for this specific pH function.

P7L8, the speciation of what?

P8L7, please justify why f(mega) and f(ferb) differ so much during initiation. A coma is missing between 0.5 and f(ferb). Also, I would keep the acronyms consistent, including their letter cases, throughout the manuscript.

L10, 'bioavailable ferric oxides' is a vague and potentially controversial term. Please provide a definition. Also justify why HCl-Fe(III) is used to represent bioavailable ferric oxides. These papers may be useful: Hyacinthe et al. 2006 Geochimica et Cosmochimica Acta 70: 4166-4180, Poulton and Canfield 2005 Chemical Geology 214: 209-221.

L26-27, I cannot follow this sentence.

L28, it appears that soil microcosms were not flushed after gas sampling. Without flushing headspace, $CO_2$ concentration builds up and consequently distorts gas diffusion. Without regular flushing, results from these incubations would misrepresent the decomposition processes in the field. Please comment.

P9L8, again, the inhibitory effect of high headspace $CO_2$ on microbial activities likely explained why the $CO_2$ level off in the microcosms. Are these results appropriate for calibrating models? Please comment.

L10, except _in_ the center organic soils.

L15, _so does_ microbial activity every year

L26, pH _increased_ with Fe(III) reduction or the _increase in pH_

Qualitatively, there is evidence to support that Fe(III) reduction competes with CH4 production. The center organic soils had the highest CH4 production, while their Fe(III) reduction was the lowest among all treatments. The authors claimed that "the impact appears less significant than expected", but I don't understand what they were expecting. Please elaborate.

P10L24-25, if microcosms in Roy Chowdhury et al. (2015) were not flushed regularly, then their results underestimated $CO_2$ production. Thus, it is not surprising that the models appeared to overestimate $CO_2$ production. Please comment.

P10L26, notice that Knoblauch et al. (2013) flushed microcosm headspace whenever $CO_2$ concentration reached 3%.

P12L6, Figure 3S, adsorption of $CO_2$ on iron oxides played a bigger role in the mineral soils than in the organic soils. Why?

L19, Table 2 should be referenced here instead of Table 1.

L20-23, I agree that higher f(labileDOC) increased model performance. But such improvement occurred mostly in samples kept at 8 degrees. For samples in -2 degrees, models with the lowest f(labileDOC) were actually the best. Please comment on the interactive effects of temperature and f(labileDOC) on $CO_2$ production.

P14L10-11, soil redox condition can also explain why mineral soils have lower pH than organic soils. Soils in reducing environments usually have high pH because reduction reactions consume protons. With a much higher water content (Table 2), organic soils are in more reducing conditions and likely have higher pH than mineral soils.

Set a hanging indent for references.

Tables should be reformatted.

---

## Referee Comment (RC3) · Anonymous Referee #3 · 22 Jul 2016

Community land model carbon nitrogen (CLM-CN) predominantly represents aerobic decomposition of SOM. In this manuscript, authors propose to include anaerobic processes in this model by integrating new experimental data for redox potential, pH, and temperature parameters from Arctic soils. This manuscript is very thorough. It's amazing to see parameterization of model with experimental data! While this work has some flaws, it is a huge step forward in closing the gap between modeling and experimental data integration. I'm impressed by the author's knowledge of biogeochemical processes in soil and effort to connect real world mechanisms to the modeling results; this is no small feat. It is clear they gave a great deal of thought to their results.

In general, I would recommend the author's provide stronger justification for determining that the most limiting factor for SOM turnover is hydrolysis of macromolecules. This both served as the foundation of this work and is continually provided as an explana-

tion to observations. While it's tough to cover all possible scenarios in soils, authors should address other potential factors that could drive the rate of SOM turnover and justify why they believe hydrolysis of macromolecules is the most limiting factor.

In the conclusion, I think it would be nice for the author's to add some suggestions for parameters/processes that could be incorporated into this model in the future or specific geochemical measurements that experimentalists should consider collecting during their studies.

—Specific comments—

P3L10-11: "…the hydrolysis and fermentation reactions have been poorly quantified." I'm not sure I follow the point being made here. Is this suggesting that hydrolysis/fermentation of SOM is poorly quantified (in general) or specifically in arctic soils?

P4L28-29: What is a "low-center polygon"? It is frequently referred to the in the text of this article, yet it is unclear to me what it is. This seems like site-specific terminology that may be worth describing. I'm not sure how many readers would know what this is. I'm also assuming the "center" sampling location is a slope since the other two are the "ridge" and "trough"?

P7L28: What do SOM3 and SOM4 represent? LabileDOC, SOM1, SOM2 and the biomass pools were described, but not SOM3 and SOM4. Furthermore, SOM4 isn't included in the fractions listed on P7L29. Is it supposed to be included in this list of fractions? If not, then why is it excluded?

P8L1-2: The turnover time of SOM3 and SOM4 are not listed – these fractions need to be better described or explain why they are excluded.

P8L7-9: Nice explanation for "back of the envelop" biomass estimation

P10L26-27: Are there other potential reasons why the rate of $CO_2$ would stabilize? Limitation of some other resource? For instance, N? Does this study have evidence to

support that rate of CO2 respiration stabilized because of hydrolysis of polymers?

P11L8: parameter Fe3= 0.02 is above the max value in the range of observed values stated on P8L14, can the authors comment on why they might need to increase this value beyond observed values to help the model better match observations for Fe(II)? Do you have any suggestions for some other parameter that should be included or other parameter values that could be altered to help achieve a better model fit, while maintaining values within experimentally observed value range?

P11L11-14: How do these model observations relate to experimental data? Is there any experimental evidence (either from your original work or other soil Fe literature) to support that as Fe3 increases there is a decrease in CH4 resulting from competition between methanogens and iron reducers? Why wouldn't this also be the case when Fe3 = 0.01?

P11L29-31: This statement contradicts L25-26. L25-26 states as pH increases, CO2 (aq) increases. L29-31 states as pH increases, CO2 (aq) decreases. Please provide an explanation.

P12L19: I keep having to look back at what "WEOC" means. I would recommend using some other terminology. Also, this sentence should reference Table 2 not Table 1.

P12L20-22: Is this comparable? The values for rapid CO2 release in Figure 4 look nearly double or triple the observed values. It appears that CO2 values for organic center at a LabileDOC = 0.02 fit the experimental data best out of all of these scenarios.

P12L29: "high center polygon trough"? I thought "center" and "trough" were two different sampling sites? Please clarify and be consistent throughout the paper. Same error P13L6.

P13L19-20: I don't follow – how do these studies demonstrate that hydrolysis of macro-molecular organics by extracellular enzymes is the rate limiting step? What about bioavailability? Limitation of some other resource?

[Figure]

P13L24-26: Please rewrite this sentence for clarity.

P13L31: "the model substantially underpredicts..." Please include a figure number.

P14L1: It could also be attributed to populations at that particular site grow more rapidly than the populations at other sites. Hard to say without a T0 measurement... I would tread lightly with this, you don't have strong experimental evidence to support this statement.

P14- first paragraph: The text says the opposite of what is demonstrated in Figure 5. Figure 5 shows the lower initial biomass results in more Ch4, FeII, pH increase, etc. Is it possible the figure legend is wrong?

P14L10-12: OK, but if the OM soils are better buffered why are there rapid changes in pH for both the observed and experimental data for OM soils? FigureS6. OM soils appear to have rapid pH changes occur sooner than mineral soils, despite buffering? Please explain.

P16L21: change "enhancing" to "enhances"

P16: Transparent science! Thanks for making your code and data available!

P17: It's unclear what a pH response and temperature response function are. Please better define. What is the reader supposed to take away from this information?

All tables and figures should be able to stand on their own. Improve caption text and add full legends (colors, symbols, and patterns defined in each figure).

-Please format Table 2.

-Figure 2 caption L5 add "as" after "such"

-Figure 5 caption text does not match figure. Legend suggests lowest initial biomass results in highest CH4. Please make full legend visible (partially covered up).

---

## Author Comment (AC3) · 2 Aug 2016

Comment 1: Soil carbon models are a critical source of uncertainty in Earth system models both due to limitation in model-data and process representation. This manuscript addresses modeling hurdles in a key process (anaerobic decomposition) missing in most soil car- bon models. This work is a timely, novel, and carefully conducted. However I am uncomfortable with the current introduction, justification, and implication presented in the manuscript. The actual results section is very strong but would suggest extensive revisions to the introduction and conclusion.

Response 1: We appreciate that the referee spent valuable time in reviewing this manuscript, and provided very constructive comments from a different perspective. As detailed below and highlighted in the revised version, we made extensive revisions

to improve the introduction, justification, and implication components as the referee suggested, with a focus on clarifying the scope of this work, and putting it in the comprehensive context of earth system model concisely. We hope the manuscript is substantially improved.

Comment 2: In general, I would suggest better connections between and within the introductory paragraphs. The main paragraphs jump around and paragraphs themselves lack coherent structure.

Response 2: This is because that we tried to avoid lengthy discussion about the context of carbon mineralization and methane production, consumption, and transport. Besides extensive revisions in the introduction as highlighted in the attached marked revised manuscript, we add the following paragraph in the end of the introduction section to concisely describe the context and limit our scope of this study, which reads:

"The carbon cycle involves coupled hydrological, geochemical, and biological processes interacting from molecular to global scales. The implicit empirical first order approach used in existing LSMs limits our understanding of the land atmosphere interaction and is a source of prediction uncertainty. To improve our understanding and reduce prediction uncertainty, we attempt to use relatively more explicit mechanistic representations developed in the reactive transport model literature (Tang et al., 2016). Even though explicit representation does not necessarily improve the match between the predictions and observations over well-tuned existing models immediately (e.g., Wieder et al. 2015; Steven et al. 2006), our approach provides a systematic means to incorporate on-going process-rich investigations to improve mechanistic representations in LSMs across scales. As a preliminary study, we constrain our scope to extending CLM-CN with minimum revision to describe anaerobic $CO_2$ and $CH_4$ production from several recent microcosm studies in this work. We discuss next steps briefly results and discussion section." We also add the paragraph in the end to describe future directions:

"As the experimental data from Roy Chowdhury et al. (2015) and Herndon et al. (2015) are shown to be invaluable, iteration of further biogeochemical modeling and new experiments that span a range of temperature, pH, redox conditions, and include detailed characterization of hydrolysis, microbial and enzymatic activities will further improve mechanistic understanding and representation of carbon mineralization and methanogenesis. The oxidation of methane, Fe(II), etc., may be equally important for simulating the carbon cycle. The biogeochemical model is typically coupled with a hydrological model to account for heterogeneity in structured soils using 3-dimensional high resolution grids. As CLM-PFLOTRAN couples CLM with a reactive transport code PFLOTRAN, these hydrobiogeochemical model developments can be directly incorporated and tested from laboratory to global scales. As we implement processes, such as gas and aqueous transport through soils and aerenchyma, explicitly representing microbial processes for carbon decomposition, incrementally, we hope the new modeling framework will be more and more useful for future investigation and land surface model developments."

Comment 3: I would also like to see a discussion/acknowledgement of other mechanisms that could influence anaerobic decomposition which were not captured by the proposed model. While not ALL processes can be included in a model, some acknowledgement of those missing processes and either how then can be incorporated into future work or how they influence current simulations are appropriate and highlight the limitations of the proposed model.

Response 3: the two paragraphs added in response 2 put the current work in the context of other processes. In responses to referee #2, we discuss describing impact of flushing on biogeochemical processes.

Comment 4: Relatedly the authors need to address how these detailed pool based kinetic models scale to well-structured heterogeneous soils. Great detail is gone into on the chemical processes governing methanogensis in the introduction but there is no discussion of how the physical structure of the soil plays into substrate and oxygen

availability and the inherent limitations to applying mechanistic pool models to highly structured soil columns. While this is a common shortcoming of soil carbon models I feel that, given the level of processes detailed covered in the model, this is critical to address and justify utilization of a pool model with such explicit process representation. Minimum the authors need to acknowledge that the scaling of known kinetics from well-mixed experiments to highly structure soil cores is a relevant open question in the field.

Response 4: As in response 2, we add discussion about using high resolution to deal with heterogeneity; in Tang et al. 2016, we actually discussed potential ways to account for oxygen limitation and even oxygen transportation and consumption; While scaling is a grand challenge, there are evidences that the parameters determined in the lab are applicable for field studies in our previous studies (e.g., Tang et al. 2013b). As a preliminary study, we feel it is better to focus on the current scope.

Comment 5: The great strength of this manuscript is the highly detailed and careful analysis of the proposed model, grounded against a data set. This was very well done and I feel should provide the backbone of a new discussion section which could be extended to suggest potential follow up experiments based on the model results. However there is no formal model-data integration nor a comparison with an adequate number of data sets to justify this as a mature component of a new CLM module, as is implied by the current introduction and discussion. If this is the manuscript that the authors want to write then I would suggest more data sets, a formal data-model integration, and demonstration of improvement to previous ratio-based models. But the current analysis would be completely appropriate in a different context which focused more on implications to future experimental designs and long term model projections. I strongly encourage the authors to carefully consider an alternative framing of this very interesting study.

Response 5: We appreciate the referee's positive comments, and very constructive advices. As discussed in response 2, this is a preliminary study with focus on mechanistic representation. We are interested in incorporating more existing data, but are

limited by lack of detailed data such as pH, Fe, organic acids, etc., in the existing studies. In the revision, we mention potential next steps and further model data iteration exercises.

–Line by line response–

Comment 6: P2L2 Actually many of the IPCC models suggest that SOC will increase in the northern latitudes due to increases in inputs (Todd-Brown et al 2014), I would suggest softening this statement to reflect the huge uncertainty in the current state of the science. Less controversial would be a statement referring to a general ramping up of the entire car- bon cycle in response to climate change, increases are expected in both primary pro- duction and decomposition. Whether the net effect will be to convert SOC to CO2/CH4 is highly debated.

Response 6: revise ". . . are widely expected to accelerate . . ." to ". . . may . . ." to soften the statement.

Comment 7: P2L2-14 I like the content of this paragraph but it needs to be re-organized. There are three distinct topics in this paragraph which would be better served breaking them up/integrating with later parts of the manuscript: a review of expected high latitude SOC vulnerability to climate change, summary of CLM-CN representation of anaerobic conditions (coupling this with a general review of aerobic decomposition would not go amiss here but that is a soft suggestion), and the comparison to lab incubations.

Response 7: we separate the paragraphs into two, and put our study in the context concisely (see marked revised manuscript).

Comment 8: P2L15 CH4 is critical not just because of it's high global warming potential but also because of it's emissions rate and residence time in the atmosphere. Please add some citations to reflect this.

Response 8: add residence time and production rate (italic) as

"Because CH4 has a 100-year global warming potential that is about 26 times greater than CO2 (Forster et al., 2007;IPCC, 2013), and an atmospheric residence time of approximately 10 years (IPCC, 2013), and methanogenesis rate can be high under favorable conditions, ..."

Comment 9: P2L19 Why is this lag critical?

Response 9: We rewrite the paragraph as follow: "Methanogenesis is widely parameterized as a fraction of carbon mineralization (Wania et al., 2013; Oleson et al., 2013; Koven et al., 2015; Cheng et al., 2013). However, the ratio of CH4 to CO2 ranges from 0.00001 to 0.5 (Wania et al., 2010; Drake et al., 2009; Bridgham et al., 2013), highlighting limitation of this simplistic empirical approach. The wide range of CH4 to CO2 ratio, also shown as the observed time lag of CH4 accumulation behind CO2 in anaerobic microcosm ranging from days to years (Knoblauch et al., 2013; Roy Chowdhury et al., 2015; Cui et al., 2015; Hoj et al., 2007; Fey et al., 2004; Jerman et al., 2009; Tang et al., 2013c), is due to different temperatures (Fey and Conrad, 2003; Hoj et al., 2007; Jerman et al., 2009; Cui et al., 2015), initial abundance of methanogens (Conrad, 1996; Knoblauch et al., 2013), and the wide range of redox buffering provided by the alternative electron acceptors (Estop-Aragonés and Blodau, 2012; Fey et al., 2004; Jerman et al., 2009; Yao et al., 1999; Conrad, 1996; Knorr and Blodau, 2009). Accordingly, methanogenesis is explicitly represented in some models (Xu et al., 2015; Grant, 1998) and the reduction of alternative electron acceptors is explicitly represented in others (Fumoto et al., 2008; Segers and Kengen, 1998; Van Bodegom et al., 2001; van Bodegom et al., 2000). However, these models do not have an aqueous phase that is essential for explicit biogeochemical calculations, e.g., pH, Eh, and thermodynamic calculations. As the free energy of methanogenesis reactions is less favorable than the reduction of O2, NO3-, Mn (IV), Fe(III), and SO42- along the redox ladder (Conrad, 1996; Bethke et al., 2011), it is important to explicitly simulate the redox condition to accurately predict methanogenesis."

to connect CH4:CO2 ratio with lag. The wide range of CH4:CO2 corresponds with the

wide range of lag time.

Comment 10: P2L25 Why is the CH4:CO2 ratio important? Maybe lead with this being a critical parameter for current models and then show how this is a dynamic response to the competing Redox ladder. I think this is where the authors are trying to go with this but it is lost in the paragraph. Could the proposed model be compared with the standard ratio model?

Response 10: As in response 9, the manuscript shows that a constant CH4:CO2 ratio approach introduces prediction uncertainty, and our mechanistic model has the potential to reduce the prediction uncertainty. We feel it to be premature to conclude whether the mechanistic model perform better than a standard ratio model. The mechanistic model is expected to work better than the empirical for these specific data sets. It is challenging to extend the comparison to other available datasets because the lack of measurements (e.g., pH, Fe, etc.) to support the mechanistic model. Our long term goal is to evaluate if a mechanistic model performs better than a standard ratio model. This is only the first step toward the long term goal.

Comment 11: P2L33 Why is an aqueous phase essential for these calculations? Soil decomposition models are implicit descriptions of carbon dynamics anyway, why do we need an explicit representation of this process? Can a micro scale process like an explicit terminal electron acceptor model be simulated on the macro scale? I would suggest placing this study in the context of the increasing number of ?explicit? soil carbon dynamics models (ex: Wieder et al 2015). These models may or may not increase the overall accuracy of predictive dynamics over a well-tuned traditional model but they can provide critical scientific insights into the process of soil decomposition. This introduction lacks this critical nuance and oversells the capabilities of the proposed model.

Response 11: The aqueous phase is essential in that pH, Eh, thermodynamics, etc., are defined in the aqueous phase. As in detailed in Response 2, we put this study in

the general context in the revision and acknowledge that explicit representation may not necessarily improve the match with the observation immediately over existing well-tuned models.

Comment 12: Sect2.2.1(and elsewhere) Please refer to model pools and other variables by name (variable) consistently throughout the manuscript, ex: organic acid pool (Ac). This reduces the need to refer back to tables/sections. Manuscripts are rarely read linearly and having to search for abbreviation definitions slows down reading.

Response 12: spell out.

Comment 13: P8L20 I applaud the authors for making their scripts available in the supplemental. Thank you.

Response 13: Thanks.

Comment 14: P12 Nice job walking through potential drivers of model-data mismatch. These provide a rich pool of candidates for future investigations. I feel that this should be the main focus of the conclusions. Given that a single data set was used and no formal model-data integration done this model is not quite ready for a full land carbon model integration as is implied by the authors in the introduction and conclusions. What IS done quite elegantly is an analysis of several representations of potential mechanisms and how they influence overall carbon mineralization in the context of a common model structure.

Response 14: Thanks for this nice evaluation. We revise the discussion and conclusion section to further strengthen these points.

Comment 15: P12 WEOC, TOTC are an unusual acronym in the field. Consider writing out the full name instead, I found myself forgetting what it stood for around here and having to go look it up. See previous comment about variable/pool references.

Response 15: spell out.

Comment 16: P13L19 This is a highly controversial statement that does not belong in the results section. While it is appropriate to highlight the relatively low amount of mineralized carbon there are several possible mechanisms for this that are unrelated to the chemical structure as suggested by this statement. Just because that is the explanation that fits into the model that is presented, it is not the only explanation (I'm thinking of various physical mechanisms like co-location and aggregate formation, as well as the substrate rarity argument ala Allson 2006). Please move to the conclusion and soften this statement considerably.

Response 16: revise "... is the rate limiting step ..." to "... could be a rate limiting step..." to soften the statement.

Comment 17: P14L17 Cite the equation reference for f(pH)

Response 17: add (Eq. 3).

Comment 18: P14L26 Given the noise generally inherent in these measurements I would hesitate to call this a substantial difference. Could you can provide error bars for the data or some kind of significance testing.

Response 18: The sentence reads "These differences translate to substantial differences in model predictions". It is model predictions not observations. We add "(Fig. S7)" to avoid confusion. Instead of showing the average with error bars, the duplicate/triplicate observations were shown for headspace $CO_2$ and $CH_4$ to demonstrate the variation. For organic acids, Fe(II), and pH, the standard deviations are too small to shown as error bars.

Comment 19: Figures: In general, would it be possible to add error/uncertainty bars to the data points in the figures? This would place the modeled sensitivity in the context of the measurement error.

Response 19: see second half of response 18.

Comment 20: P16L5-7 WHAM is an aqueous pool model, claiming that there is no

needed modifications when applying it to a well structured soil column seems a bit of a stretch.

Response 20: Windermere Humic Aqueous Model (WHAM) appears to be an aqueous model from its name. In fact, it treats the binding sites in soil organic matter as surface sites, and include other minerals such as Fe(OH)3, Al(OH)3, etc. In our implementation, we use surface sites to simulate organic matter. Namely, it does include solid phase, which is critical for soils.

Comment 22: Table 2 Formatting needs to be fixed for the table entries and I would suggest replacing TOTC with Total Organic Carbon and WEOC with Water Extractable Organic Carbon. OC is a common enough abbr. that it could be used here without explanation but TOTC and WEOC are not.

Response 22: Spell out TOTC and WEOC.

Comment 23: Figure and Table captions: Figure and table captions need to be able to stand alone in the manuscript, people will often scan the figures to get a sense of the results of the manuscript. Please expand the figure captions to more fully reflect the conclusions being illustrated here, this is particularly needed for the supplemental figures.

Response 23: This is improved as shown in the marked revised version of the manuscript and supplement. Please see the enclosed documents for details.

===================== Todd-Brown, K. E. O., Randerson, J. T., Hopkins, F., Arora, V., Hajima, T., Jones, C., Shevliakova, E., Tjiputra, J., Volodin, E., Wu, T., Zhang, Q. and Allison, S. D.: Changes in soil organic carbon storage predicted by Earth system models during the 21st century, Biogeosciences, 11(8), 2341-2356, 10.5194/bg-11-2341-2014, 2014. Wieder, W. R., Allison, S. D., Davidson, E. A., Georgiou, K., Hararuk, O., He, Y., Hop-kins, F., Luo, Y., Smith, M. J., Sulman, B., Todd-Brown, K., Wang, Y.-P., Xia, J. and Xu, X.: Explicitly representing soil microbial processes in Earth system

models, Global Biogeochem. Cycles, 29(10), 1782-1800, 10.1002/2015GB005188, 2015. Steven D. Allison and Allison, S. D.: Brown Ground: A soil carbon analogue for the green world hypothesis?, Am. Nat., 167(5), 619-627, 10.1086/503443, 2006.

The marked revised manuscript and supplement are enclosed as supplement in this response.

Please also note the supplement to this comment:
http://www.biogeosciences-discuss.net/bg-2016-207/bg-2016-207-AC3-supplement.zip

---

## Author Response (AR1)

**Responses to the Associate Editor**

Dear Dr. Keenan,

Thank you very much for handling the review of our manuscript. We appreciate your extra effort in reaching out to three referees and securing quality review comments. We are very pleased by the overall positive, encouraging and constructive comments. We thank the three referees for their time and help. As shown in the point-to-point responses to the three referees, we have made extensive revisions to address every of the comments, with the major ones summarized as below:

Referee #1 expressed discomfort with the introduction, justification, and implication, and suggested better connection between the introductory paragraphs, and alternative framing of the results and discussion. We rewrote the introduction section and added discussion in the final section to clarify the scope of this work, and put it in the comprehensive context of earth system model concisely.

Referee #2 was concerned about the impact of head space trace gas accumulation on carbon mineralization, and potential limitation of the data from microcosms without flushing in reflecting field conditions. We conducted additional numerical experiments using biogeochemical model to evaluate the impact. We demonstrated that the impact is complicate (highly nonlinear as the controlling coupled processes are highly nonlinear), and increasingly mechanistic model are useful in help us understand the processes, with the hope to improve our predictability.

Referee #3 recommended that we provide stronger justification for determining that the most limiting factor for SOM turnover is hydrolysis of macromolecules. We agree that hydrolysis is a limiting factor. We also agree with referee #1 that it is controversial to state that hydrolysis is the most limiting factor. As a result, we try to be balanced and discuss about possible new data needs to better understand and quantify hydrolysis.

With those revisions, the manuscript has been substantially improved. We hope the revised manuscript is acceptable for publication Biogeosciences and will be helpful for both modeling and experimental research. We appreciate you and the three referees for help. Should you have any question, please feel free to contact us.

**Response to Referee #1**

**Comment 1:** Soil carbon models are a critical source of uncertainty in Earth system models both due to limitation in model-data and process representation. This manuscript addresses modeling hurdles in a key process (anaerobic decomposition) missing in most soil car- bon models. This work is a timely, novel, and carefully conducted. However I am uncomfortable with the current introduction, justification, and implication presented in the manuscript. The actual results section is very strong but would suggest extensive revisions to the introduction and conclusion.

**Response 1:** We appreciate that the referee spent valuable time in reviewing this manuscript, and provided very constructive comments from a different perspective. As detailed below and highlighted in the revised version, we made extensive revisions to improve the introduction, justification, and implication components as the referee suggested, with a focus on clarifying the scope of this work, and putting it in the comprehensive context of earth system model concisely. We hope the manuscript is substantially improved.

**Comment 2:** In general, I would suggest better connections between and within the introductory paragraphs. The main paragraphs jump around and paragraphs themselves lack coherent structure.

**Response 2:** This is because that we tried to avoid lengthy discussion about the context of carbon mineralization and methane production, consumption, and transport. We add the following paragraph in the end of the introduction section to concisely describe the context and limit our scope of this study, which reads:

"The carbon cycle involves coupled hydrological, geochemical, and biological processes interacting from molecular to global scales. The implicit empirical first order approach used in existing LSMs limits our understanding of the land atmosphere interaction and is a source of prediction uncertainty. To improve our understanding and reduce prediction uncertainty, we attempt to use relatively more explicit mechanistic representations developed in the reactive transport model literature (Tang et al., 2016). Even though explicit representation does not necessarily improve the match between the predictions and observations over well-tuned existing models immediately (e.g., Wieder et al. 2015; Steven et al. 2006), our approach provides a systematic means to incorporate on-going process-rich investigations to improve mechanistic representations in LSMs across scales. As a preliminary study, we constrain our scope to extending CLM-CN with minimum revision to describe anaerobic $CO_2$ and $CH_4$ production from several recent microcosm studies in this work. We discuss next steps briefly results and discussion section."

We also add two paragraphs in the end of the results and discussion section to describe the implications as:

"Another following up task is to assess the model in field studies using CLM-PFLOTRAN. This can be done incrementally, i.e., add/remove reactions one at a time without source code modifications. CLM-PFLOTRAN currently uses CLM4.5 vertical resolved grid, The resolution can be adjusted, possibly in three dimension, to reflect the heterogeneity of any structural soil column to account for the limitation of electron donors and electron acceptors at individual locations. As we gradually implement more and more processes, such as gas and aqueous transport, through soils and aerenchyma, explicitly representing microbial processes for carbon decomposition, we hope the new framework will be useful for future investigation and model developments."

Note that the first paragraph is in response to the comments about headspace flushing raised by the second referee.

**Comment 3**: I would also like to see a discussion/acknowledgement of other mechanisms that could influence anaerobic decomposition which were not captured by the proposed model. While not ALL processes can be included in a model, some acknowledgement of those missing processes and either how then can be incorporated into future work or how they influence current simulations are appropriate and highlight the limitations of the proposed model.

**Response 3:** as in response 2, we mention the hydrological and biological processes across scales in general; for specific processes, we discuss about mimic impact of flushing/boundary condition/pressure, as well as increase the grid resolution to simulate the heterogeneous structured soil columns. And we talk about potential next steps.

**Comment 4**: Relatedly the authors need to address how these detailed pool based kinetic models scale to well-structured heterogeneous soils. Great detail is gone into on the chemical processes governing methanogensis in the introduction but there is no discussion of how the physical structure of the soil plays into substrate and oxygen availability and the inherent limitations to applying mechanistic pool models to highly structured soil columns. While this is a common shortcoming of soil carbon models I feel that, given the level of processes detailed covered in the model, this is critical to address and justify utilization of a pool model with such explicit process representation. Minimum the authors need to acknowledge that the scaling of known kinetics from well-mixed experiments to highly structure soil cores is a relevant open question in the field.

**Response 4:** As in response 2, we add discussion about using high resolution to deal with heterogeneity; in Tang et al. 2016, we actually discussed potential ways to account for oxygen limitation and even oxygen transportation and consumption; While scaling is a grand challenge, there are evidences that the parameters determined in the lab are applicable for field studies in our previous studies (e.g., Tang et al. 2013b). As a preliminary study, we feel it is better to focus on the current scope.

**Comment 5**: The great strength of this manuscript is the highly detailed and careful analysis of the proposed model, grounded against a data set. This was very well done and I feel should provide the backbone of a new discussion section which could be extended to suggest potential follow up experiments based on the model results. However there is no formal model-data integration nor a comparison with an adequate number of data sets to justify this as a mature component of a new CLM module, as is implied by the current introduction and discussion. If this is the manuscript that the authors want to write then I would suggest more data sets, a formal data-model integration, and demonstration of improvement to previous ratio-based models. But the current analysis would be completely appropriate in a different context which focused more on implications to future experimental designs and long term model projections. I strongly encourage the authors to carefully consider an alternative framing of this very interesting study.

**Response 5**: We appreciate the referee's positive comments, and very constructive advices. As discussed in response 2, this is a preliminary study with focus on mechanistic representation. We are interested in incorporating more existing data, but are limited by lack of detailed data such as pH, Fe, organic acids, etc., in the existing studies. In the revision, we mention potential next steps and further model data iteration exercises.

--Line by line response--

**Comment 6:** P2L2 Actually many of the IPCC models suggest that SOC will increase in the northern latitudes due to increases in inputs (Todd-Brown et al 2014), I would suggest softening this statement to reflect the huge uncertainty in the current state of the science. Less controversial would be a statement referring to a general ramping up of the entire car- bon cycle in response to climate change, increases are expected in both primary pro- duction and decomposition. Whether the net effect will be to convert SOC to CO2/CH4 is highly debated.

**Response 6:** revise "… are widely expected to accelerate …" to "… may …" to soften the statement.

**Comment 7:** P2L2-14 I like the content of this paragraph but it needs to be re-organized. There are three distinct topics in this paragraph which would be better served breaking them up/integrating with later parts of the manuscript: a review of expected high latitude SOC vulnerability to climate change, summary of CLM-CN representation of anaerobic conditions (coupling this with a general review of aerobic decomposition would not go amiss here but that is a soft suggestion), and the comparison to lab incubations.

**Response 7:** we separate the paragraphs into two, and put our study in the context concisely (see marked revised manuscript).

**Comment 8:** P2L15 CH4 is critical not just because of it's high global warming potential but also because of it's emissions rate and residence time in the atmosphere. Please add some citations to reflect this.

**Response 8:** add residence time and production rate (*italic*) as

"Because $CH_4$ has a 100-year global warming potential that is about 26 times greater than $CO_2$ (Forster et al., 2007;IPCC, 2013), *and an atmospheric residence time of approximately 10 years (IPCC, 2013), and methanogenesis rate can be high under favorable conditions,* …"

**Comment 9:** P2L19 Why is this lag critical?

**Response 9:**  add

"The implication is that a first order rate (including constant $CO_2$ $CH_4$ ratio parameterization) overpredicts $CH_4$ production rate before methanogenosis initiation and underpredicts $CH_4$ production rate afterwards, and the uncertain lag time introduces large uncertainty in $CH_4$ production prediction."

**Comment 10:** P2L25 Why is the CH4:CO2 ratio important? Maybe lead with this being a critical parameter for current models and then show how this is a dynamic response to the competing Redox ladder. I think this is where the authors are trying to go with this but it is lost in the paragraph. Could the proposed model be compared with the standard ratio model?

**Response 10:**  As in response 9, the manuscript shows that a constant CH4:CO2 ratio approach introduces prediction uncertainty, and our mechanistic model has the potential to reduce the prediction uncertainty. We feel it to be premature to conclude whether the mechanistic model perform better than a standard ratio model. The mechanistic model is expected to work better than the empirical for these specific data sets. It is challenging to extend the comparison to other available datasets because the lack of measurements (e.g., pH, Fe, etc.) to support the mechanistic model.  Our long term goal is to evaluate if a mechanistic model performs better than a standard ratio model. This is only the first step toward the long term goal.

**Comment 11:** P2L33 Why is an aqueous phase essential for these calculations? Soil decomposition models are implicit descriptions of carbon dynamics anyway, why do we need an explicit representation of this process? Can a micro scale process like an explicit terminal electron acceptor model be simulated on the macro scale? I would suggest placing this study in the context of the increasing number of ?explicit? soil carbon dynamics models  (ex: Wieder et al 2015). These models may or may not increase the overall accuracy of predictive dynamics over a well-tuned traditional model but they can provide critical scientific insights into the process of soil decomposition. This introduction lacks this critical nuance and oversells the capabilities of the proposed model.

**Response 11:** The aqueous phase is essential in that pH, Eh, thermodynamics, etc., are defined in the aqueous phase. As in detailed in Response 2, we put this study in the general context in

the revision and acknowledge that explicit representation may not necessarily improve the match with the observation immediately over existing well-tuned models.

**Comment 12**: Sect2.2.1(and elsewhere) Please refer to model pools and other variables by name (variable) consistently throughout the manuscript, ex: organic acid pool (Ac). This reduces the need to refer back to tables/sections. Manuscripts are rarely read linearly and having to search for abbreviation definitions slows down reading.

**Response 12:** spell out.

**Comment 13:** P8L20 I applaud the authors for making their scripts available in the supplemental. Thank you.

**Response 13:** Thanks.

**Comment 14:** P12 Nice job walking through potential drivers of model-data mismatch. These provide a rich pool of candidates for future investigations. I feel that this should be the main focus of the conclusions. Given that a single data set was used and no formal model-data integration done this model is not quite ready for a full land carbon model integration as is implied by the authors in the introduction and conclusions. What IS done quite elegantly is an analysis of several representations of potential mechanisms and how they influence overall carbon mineralization in the context of a common model structure.

**Response 14:** Thanks for this nice evaluation. We revise the discussion and conclusion section to further strengthen these points.

**Comment 15:** P12 WEOC, TOTC are an unusual acronym in the field. Consider writing out the full name instead, I found myself forgetting what it stood for around here and having to go look it up. See previous comment about variable/pool references.

**Response 15**: spell out.

**Comment 16:** P13L19 This is a highly controversial statement that does not belong in the results section. While it is appropriate to highlight the relatively low amount of mineralized carbon there are several possible mechanisms for this that are unrelated to the chemical structure as suggested by this statement. Just because that is the explanation that fits into the model that is presented, it is not the only explanation (I'm thinking of various physical mechanisms like co-location and aggregate formation, as well as the substrate rarity argument ala Allson 2006). Please move to the conclusion and soften this statement considerably.

**Response 16:** revise "… is the rate limiting step …" to "… could be a rate limiting step…" to soften the statement.

**Comment 17:** P14L17 Cite the equation reference for f(pH)

**Response 17:** add (Eq. 3).

**Comment 18:** P14L26 Given the noise generally inherent in these measurements I would hesitate to call this a substantial difference. Could you can provide error bars for the data or some kind of significance testing.

**Response 18**: The sentence reads "These differences translate to substantial differences in model predictions". It is model predictions not observations. We add "(Fig. S7)" to avoid confusion. Instead of showing the average with error bars, the duplicate/triplicate observations were shown for headspace $CO_2$ and $CH_4$ to demonstrate the variation. For organic acids, Fe(II), and pH, the standard deviations are too small to shown as error bars.

**Comment 19:** Figures: In general, would it be possible to add error/uncertainty bars to the data points in the figures? This would place the modeled sensitivity in the context of the measurement error.

**Response 19:** see second half of response 18.

**Comment 20:** P16L5-7 WHAM is an aqueous pool model, claiming that there is no needed modifications when applying it to a well structured soil column seems a bit of a stretch.

**Response 20**: Windermere Humic Aqueous Model (WHAM) appears to be an aqueous model from its name. In fact, it treats the binding sites in soil organic matter as surface sites, and include other minerals such as $Fe(OH)_3$, $Al(OH)_3$, etc. In our implementation, we use surface sites to simulate organic matter. Namely, it does include solid phase, which is critical for soils.

**Comment 21:** Table 2 Formatting needs to be fixed for the table entries and I would suggest replacing TOTC with Total Organic Carbon and WEOC with Water Extractable Organic Carbon. OC is a common enough abbr. that it could be used here without explanation but TOTC and WEOC are not.

**Response 21**:  Spell out TOTC and WEOC.

**Comment 22:** Figure and Table captions: Figure and table captions need to be able to stand alone in the manuscript, people will often scan the figures to get a sense of the results of the manuscript. Please expand the figure captions to more fully reflect the conclusions being illustrated here, this is particularly needed for the supplemental figures.

**Response 22:**  This is improved as shown in the marked revised version of the manuscript and supplement. Please see the enclosed documents for details.

=====================

Todd-Brown, K. E. O., Randerson, J. T., Hopkins, F., Arora, V., Hajima, T., Jones, C., Shevliakova, E., Tjiputra, J., Volodin, E., Wu, T., Zhang, Q. and Allison, S. D.: Changes in soil organic carbon storage predicted by Earth system models during the 21st century, Biogeosciences, 11(8), 2341-2356, 10.5194/bg-11-2341-2014, 2014.

Wieder, W. R., Allison, S. D., Davidson, E. A., Georgiou, K., Hararuk, O., He, Y., Hop-kins, F., Luo, Y., Smith, M. J., Sulman, B., Todd-Brown, K., Wang, Y.-P., Xia, J. and Xu, X.: Explicitly representing soil microbial processes in Earth system models, Global Biogeochem. Cycles, 29(10), 1782-1800, 10.1002/2015GB005188, 2015.

Steven D. Allison and Allison, S. D.: Brown Ground: A soil carbon analogue for the green world hypothesis?, Am. Nat., 167(5), 619-627, 10.1086/503443, 2006.

**Response to Referee #2**

**Comment 1:** The manuscript proposes a new module to the CLM-CN. It attempts to address a key limitation of existing land surface models in that the anaerobic decomposition is poorly represented. This work is novel and quite thorough. Most of my comments are technical except that I am not 100% comfortable with the testing dataset. It appears that the incubation experiment did not flush the headspace of microcosms. (This manuscript and the original publication both neglected to discuss flushing). It is well known that the accumulation of $CO_2$ and other trace gases in headspace distorts gas diffusion and greatly impacts decomposition. The suitability of the testing dataset is worth a thorough discussion.

**Response 1:** We appreciate the positive comments and the reasonable concerns this referee provided. Thanks for your time and consideration.

The major concern is the impact of head space trace gas accumulation on carbon mineralization, and potential limitation of the data from microcosms without flushing in reflecting field conditions. As traditional models (e.g., Knoblauch et al. 2013) do not explicitly account for the tracer gas accumulation impact on mineralization and methanogenesis, it is necessary to mimic the field conditions in the experiments. However, it is not clear that what level of flushing reflect the field conditions. Biogeochemical models like ours explicitly account for the impact of tracer gas accumulation in the headspace. They can describe the observations under various conditions, ranging from open to close system including various levels of flushing. The experimental data in this work were collected in well-defined conditions, and are valuable in quantifying these underlying processes, which is the focus of this work.

To address this concern and acknowledge the limitation, we add numerical experiments with 10 and 100 times headspace volumes. In the results and discussion section, we add the following:

"3.2.9 Predicted impact of headspace gas accumulation

Knoblauch et al. (2013) and Yang et al. (2016) flushed the headspace of the microcosms while Roy Chowdhury et al. (2015) and Herndon et al. (2015) did not. The field conditions are likely somewhere between an open system and a closed system because neither the atmospheric pressure nor the hydrostatic pressure is constant, and the produced CO2 and CH4 are not always free to release to the atmosphere. To assess the impact of CO2 accumulation in the headspace on the soil carbon mineralization and methanogenesis, we conduct numerical experiments with 10 and 100 times the headspace volume of the experimental values. With increased headspace volume, the headspace and aqueous CO2 concentrations are predicted to decrease (Fig. S11 f1-6, g1-6), and the pH increase is predicted to slow down. As a result, the biogeochemical reaction rates are generally slower (Fig. S11e1-6). Eventually, the predicted total CO2 and CH4 production generally decrease with lower headspace CO2 concentration (Fig. S11a1-6,b1-6). However, the impact on CO2 production is very small for the organic soils in the trough and ridge location, and the CO2 production is predicted to increase with decrease in headspace CO2 concentration for the organic center soils. Because of the complicated nonlinear relationships in the biogeochemical processes, the impact of headspace gas accumulation on carbon mineralization and methanogenesis is not linear. While it is debatable

that which experimental conditions (flush the headspace or not) reflect the field conditions, biogeochemical models like ours provide a mechanistic method to account for this impact by using boundary conditions that reflect the reality. Additional targeted experiments and mechanistic models are necessary to better understand the impact under different conditions, and develop representations that reflect field conditions."

[Figure]

**Figure S1: Impact of headspace volume on predictions: increase in headspace volume results in decrease in headspace and aqueous CO₂ concentration, slower pH increase and biogeochemical reaction rates, and generally less CO₂ and CH₄ production prediction. As an exception, predicted CO₂ production is increases with increasing headspace volume for the center oganic soils. The impact is not linear as the underlying biogeochemical processes are nonlinear. See Fig.2 caption for more description about the model and experimental parameters.**

**Comment 2:** I am very intrigued by this model, especially its addition of geochemical components. From an empirical perspective, I agree that redox dynamics and the turnover of pH and single substrates are all key in understanding anoxic decomposition. However, it is still unclear whether adding these processes make sense in modeling decomposition and trace gases production. Personally, I would like to see a comparison between the new model and CLM4ME/CLM-CN and evaluate whether the new model offers meaningful improvement. Other data sources could also be used to calibrate the model. These and other future directions should be discussed in details in the conclusion.

**Response 2**: We agree that it is debatable that whether adding these processes makes sense, and more work is needed to check if these improvements work. These are in our following work therefore are not included in this study. In our revision, we add in the end of the introduction section to better define our context and scope:

"The carbon cycle involves coupled hydrological, geochemical, and biological processes interacting from molecular to global scales. The implicit empirical first order approach used in existing LSMs limits our understanding of the land atmosphere interaction and is a source of prediction uncertainty. To improve our understanding and reduce prediction uncertainty, we attempt to use relatively more explicit mechanistic representations developed in the reactive transport model literature (Tang et al., 2016). Even though explicit representation does not necessarily improve the match between the predictions and observations over well-tuned existing models immediately (e.g., Wieder et al. 2015; Steven et al. 2006), our approach provides a systematic means to incorporate on-going process-rich investigations to improve mechanistic representations in LSMs across scales. As a preliminary study, we constrain our scope to extending CLM-CN with minimum revision to describe anaerobic $CO_2$ and $CH_4$ production from several recent microcosm studies in this work. We discuss next steps briefly results and discussion section."

**Comment 3:** Title, would it be better to say biogeochemical modeling or A biogeochemical model of CO2 and CH4 production?

**Response 3:** revise to "biogeochemical modeling of …"

**Comment 4:** When citing references, please add a space after each semicolon.

**Response 4:** Done. These were due to the automatic formatting using endnote.

**Comment 5:** P3L2, there is a disconnect between the topic sentence and the following text. The topic sentence introduces 'simple substrates' and their importance in modeling CH4, while the following sentences switched the focus away from simple substrates. This paragraph can be reorganized to improve the flow of thought.

**Response 5:** add between the first two sentences "Instead, they are typically lumped together as dissolved organic matter (DOM) or low-molecular-weight organic carbon (LMWOC)." to connect the sentences.

**Comment 6:** P3L8, lignin should not be classified as polysaccharides.

**Response 6:** Removed "lignin" from the list.

**Comment 7:** P3L14, acetate and H2 have been added to models?

**Response 7:** revise to "representations of acetate and $H_2$ have been added to models…"

**Comment 8:** When introducing soil pH and its role in LSMs, it is worth noting that soil pH changes rapidly as a result of redox reactions. For instance, iron(III) reduction consumes protons and usually increases soil pH.

**Response 8:** the paragraph is revised from

"In addition to electron acceptors and substrates, SOM turnover is also sensitive to soil pH. Most methanogens grow over a relatively narrow pH range (6-8), while some adapt to acidic or basic environments (Garcia et al., 2000; Van Kessel and Russell, 1996; Wang et al., 1993; Sowers et al., 1984; Rivkina et al., 2007; Hao et al., 2012; Kotsyurbenko et al., 2004; Kotsyurbenko et al., 2007). The pH response functions in LSMs are empirical and vary substantially (Xu et al., 2016b). Because of the large buffer capacity, soil pH is often fixed in LSMs (Oleson et al., 2013; Tian et al., 2010). But in reality, pH does change 1-2 logarithmic units in laboratory incubations (Xu et al., 2015; Roy Chowdhury et al., 2015; Peters and Conrad, 1996; Drake et al., 2015) and in the field, where it can vary significantly through the soil profile and along topographic and vegetation gradients (Cao et al., 1995; Van Bodegom et al., 2001; Lipson et al., 2013b). pH is calculated using soil acidity and soil buffer capacity (Van Bodegom et al., 2001) or as a function of acetate concentration (Xu et al., 2015). It is desirable to use a geochemical model to describe pH evolution mechanistically."

to

"In addition to electron acceptors and substrates, SOM turnover is also sensitive to soil pH. Most methanogens grow over a relatively narrow pH range (6-8), while some adapt to acidic or basic environments (Garcia et al., 2000; Van Kessel and Russell, 1996; Wang et al., 1993; Sowers et al., 1984; Rivkina et al., 2007; Hao et al., 2012; Kotsyurbenko et al., 2004; Kotsyurbenko et al., 2007). Soil pH does change 1-2 logarithmic units in laboratory incubations (Xu et al., 2015; Roy Chowdhury et al., 2015; Peters and Conrad, 1996; Drake et al., 2015). It can vary significantly through the soil profile and along topographic and vegetation gradients in the field (Cao et al., 1995; Van Bodegom et al., 2001; Lipson et al., 2013b). However, soil pH is often fixed in LSMs (Oleson et al., 2013; Tian et al., 2010). pH is calculated using soil acidity and soil buffering capacity (Van Bodegom et al., 2001) or as a function of acetate concentration (Xu et al., 2015). It is desirable to use a geochemical model to describe pH evolution mechanistically. The pH

response functions (reaction rate factor as a function of pH) in LSMs are empirical and vary substantially (Xu et al., 2016b).  Assessing the efficacy of  these functions is needed to better present pH impact on carbon mineralization and methanogenesis."

We do not feel comfortable to say soil pH change rapidly. We delete that sentence "because of the large buffering capacity, " as well.

**Comment 9:** P3L25, use pH buffering capacity instead of pH buffer capacity. It would also be useful to define 'pH response functions'.

**Response 9:** revise "buffer" to "buffering", and add "(reaction rate factor as a function of pH)" after pH response function as described above.

**Comment 10:** P3L26, logarithmic

**Response: 10:** revise from **"**logarithm" to "logarithmic" as above

**Comment 11:** P4L4, the last sentence is rather weak and does not connect with the next paragraph. A better transition is needed to highlight why it is necessary to compare temperature response functions.

**Response 11:** add (italic) to the last sentence as

"*To reduce prediction uncertainty for carbon mineralization and methanogenesis under various temperatures,* the temperature response functions need to be assessed  as well."

**Comment 12:** In 2.2.2, would it be possible to discuss how iron reduction and methanogenesis interact in the model? I can see that both of them reply on mD and kD, but what determines the partitioning of electron donors between these two processes? These details would be valuable to interpret the results later on (P9L28-35).

**Response 12:** add

"In this model, iron reducers and methanogens interact in different ways under various conditions. When the electron donors (acetate and $H_2$) are abundant, iron reducers grow faster than methanogens when Fe(III) is not limiting (depending on the $Fe(OH)_{3a}$ surface sites and iron reducers population), i.e., iron reducers have a short doubling time than methanogens. When the electron donors are limiting, iron reducers are expected to outcompete methanogens, depending on the half saturation (substrate affinity) values. The model also accounts for the thermodynamics. However, it does not account for possible different responses to temperatures and pH for iron reducers and methanogens."

**Comment 13:**  P6L26, Riley et al. (2011) in fact cited Meng et al. (2012) Biogeosciences, 9, 2793–2819 for this specific pH function.

**Response 13:** add the citation/reference.

**Comment 14:** P7L8, the speciation of what?

**Response 14:** add "$CO_2$, $CH_4$, $H_2$, Fe, etc."

**Comment 15:** P8L7, please justify why f(mega) and f(ferb) differ so much during initiation. A coma is missing between 0.5 and f(ferb). Also, I would keep the acronyms consistent, including their letter cases, throughout the manuscript.

**Response 15:** revise "$f_{mega} = f_{megh} = 0.5 f_{ferb} = 10^{-6}$" to "$f_{bio} = 10^{-6}$, $f_{MeGA} = f_{MeGH} = f_{bio}$, and $f_{FeRB} = 2 f_{bio}$". som1, som2, som3, megh, mega, and ferb in the subscripts are changed to consistent forms as normal text throughout the manuscript. $f_{FeRB}$ is twice $f_{MeGA}$ or $f_{MeGH}$ because FeRB has only one group while methanogens have two groups.

**Comment 16:** P8L10, 'bioavailable ferric oxides' is a vague and potentially controversial term. Please provide a definition. Also justify why HCl-Fe(III) is used to represent bioavailable ferric oxides. These papers may be useful: Hyacinthe et al. 2006 Geochimica et Cosmochimica Acta 70: 4166-4180, Poulton and Canfield 2005 Chemical Geology 214:

209-221.

**Response 16**: modify the sentence from "we start with $f_{fe3} = 0.005$" to "While bioavailable Fe(III) in soils is not well defined (e.g., Hyacinthe et al. 2006; Poulton and Canfield 2005), we start with $f_{fe3} = 0.005$ and evaluate the impact with a range of values." and add the references."

**Comment 17:** P8L26-27, I cannot follow this sentence.

**Response 17:** the sentence

"The overall observations appear to separate between the soil horizons (organic vs. mineral soils) rather than among the microtopographic locations (center, ridge, and trough) of ice-wedge polygons."

is rewritten as

"The variations in the overall observations appear to be better explained by the differences between the soil horizons (organic vs. mineral soils) than among the microtopographic locations (center, ridge, and trough) of ice-wedge polygons."

**Comment 18:** P8L28, it appears that soil microcosms were not flushed after gas sampling. Without flushing headspace, CO2 concentration builds up and consequently distorts gas diffusion. Without regular flushing, results from these incubations would misrepresent the decomposition processes in the field. Please comment.

**Response 18:** as in response 1 in the beginning, our model mechanistically describe the impact of headspace gas composition on mineralization and methanogenesis, therefore, better represent these processes; the field conditions can be simulated by using appropriate boundary conditions.

**Comment 19:** P9L8, again, the inhibitory effect of high headspace $CO_2$ on microbial activities likely explained why the CO2 level off in the microcosms. Are these results appropriate for calibrating models? Please comment.

**Response 19:** as above, our results are generally applicable.

**Comment 20:** P9L10, except in the center organic soils.

**Response 20:** add comma.

**Comment 21:** P9L15, so does microbial activity every year
**Response 21:** add "so does".

**Comment 22:** P9L26, pH increased with Fe(III) reduction or the increase in pH
**Response 22**: revise from "the pH increase" to "the increase in pH".

**Comment 23:** Qualitatively, there is evidence to support that Fe(III) reduction competes with CH4 production. The center organic soils had the highest CH4 production, while their Fe(III) reduction was the lowest among all treatments. The authors claimed that "the impact appears less significant than expected", but I don't understand what they were expecting. Please elaborate.
**Response 23**: This is in relative to the belief of strict thermodynamic/redox ladder where methanogenesis does not occur until all alternative electron acceptors that are more favorable than $CO_2$ are exhausted. We add "(e.g., complete inhibition until bioavailable Fe(III) is exhausted )" to "the impact appears less significant than expected".

**Comment 24:** P10L24-25, if microcosms in Roy Chowdhury et al. (2015) were not flushed regularly, then their results underestimated CO2 production. Thus, it is not surprising that the models appeared to overestimate CO2 production. Please comment.

**Response 24:** as addressed in the beginning, our model mechanistically describe the impact of headspace gas composition on mineralization and methanogenesis, therefore, better represent these processes; the field conditions can be simulated by using appropriate boundary conditions.

**Comment 25:** P10L26, notice that Knoblauch et al. (2013) flushed microcosm headspace whenever $CO_2$ concentration reached 3%.

**Response 25:** see response 1.

**Comment 26:** P12L6, Figure 3S, adsorption of $CO_2$ on iron oxides played a bigger role in the mineral soils than in the organic soils. Why?

**Response 26:** This is because of relatively higher $Fe(OH)_3a$ in the mineral soils than in the organic soils.

**Comment 27:** P12L19, Table 2 should be referenced here instead of Table 1.

**Response 27:** corrected. Thanks for catching this mistake.

**Comment 28:** P12L20-23, I agree that higher f(labileDOC) increased model performance. But such improvement occurred mostly in samples kept at 8 degrees. For samples in -2 degrees, models with the lowest f(labileDOC) were actually the best. Please comment on the interactive effects of temperature and f(labileDOC) on $CO_2$ production.

**Response 28:** as describe in section 3.1, the observed temperature response is diverse and challenging to explain. We do not feel comfortable in assessing the predictions at -2 °C.

**Comment 29:** P14L10-11, soil redox condition can also explain why mineral soils have lower pH than organic soils. Soils in reducing environments usually have high pH because reduction reactions consume protons. With a much higher water content (Table 2), organic soils are in more reducing conditions and likely have higher pH than mineral soils.

**Response 29:** add "… and/or more reducing condition in the organic soils as reduction reactions typically consume protons."

**Comment 30:** Set a hanging indent for references.

**Response 30:** Done

**Comment 31:** Tables should be reformatted.

**Response 31:** done.

**Responses to Referee #3**

**Comment 1:** Community land model carbon nitrogen (CLM-CN) predominantly represents aerobic decomposition of SOM. In this manuscript, authors propose to include anaerobic processes in this model by integrating new experimental data for redox potential, pH, and temperature parameters from Arctic soils. This manuscript is very thorough. It's amazing to see parameterization of model with experimental data! While this work has some flaws, it is a huge step forward in closing the gap between modeling and experimental data integration. I'm impressed by the author's knowledge of biogeochemical processes in soil and effort to connect real world mechanisms to the modeling results; this is no small feat. It is clear they gave a great deal of thought to their results.

**Response 1:** Many thanks for the compliments and very nice constructive comments.

**Comment 2**: In general, I would recommend the author's provide stronger justification for determining that the most limiting factor for SOM turnover is hydrolysis of macromolecules. This both served as the foundation of this work and is continually provided as an explanation to observations. While it's tough to cover all possible scenarios in soils, authors should address other potential factors that could drive the rate of SOM turnover and justify why they believe hydrolysis of macromolecules is the most limiting factor.

**Response 2**: We agree that hydrolysis is a limiting factor. We also agree with referee #1 that it is controversial to state that hydrolysis is the most limiting factor. At least, the evidence from the data referred in this work does not unequivocally substantiate the statement. As a result, we try to be balanced and discuss about possible new data needs to better understand and quantify hydrolysis.

All the referees comment on the need to mention other factors. In response, we make revisions to clarify the scope of this work, to put our work in the context of comprehensive hydrologic, geochemical and biologic processes that control soil carbon mineralization, and describe using 3-D high resolution grids to account for heterogeneity, and CLM-PFLOTRAN to use reactive transport models to improve the mechanistic representation in land surface models. Please see response to other referees for more details.

**Comment 3**. In the conclusion, I think it would be nice for the author's to add some suggestions for parameters/processes that could be incorporated into this model in the future or specific geochemical measurements that experimentalists should consider collecting during their studies.

**Response 3**: These are nice suggestions. As mentioned in Response 2, we add discussions about next steps.

—Specific comments—

**Comment 4**: P3L10-11: "… the hydrolysis and fermentation reactions have been poorly quantified." I'm not sure I follow the point being made here. Is this suggesting that hydrolysis/fermentation of SOM is poorly quantified (in general) or specifically in arctic soils?

**Response 4**: add "represented and quantified in Arctic as well as temperate and tropical soils" to clarify the point.

**Comment 5**: P4L28-29: What is a "low-center polygon"? It is frequently referred to the in the text of this article, yet it is unclear to me what it is. This seems like site-specific terminology that may be worth describing. I'm not sure how many readers would know what this is. I'm also assuming the "center" sampling location is a slope since the other two are the "ridge" and "trough"?

**Response 5**: add "(a typical arctic geographic feature in the low lands with soils surround by ice wedges, see cited references for more information)".

**Comment 6**: P7L28: What do SOM3 and SOM4 represent? LabileDOC, SOM1, SOM2 and the biomass pools were described, but not SOM3 and SOM4. Furthermore, SOM4 isn't included in the fractions listed on P7L29. Is it supposed to be included in this list of fractions? If not, then why is it excluded?

**Response 6**: SOM3 and SOM4 are like SOM1 and SOM2, two additional soil organic matter pools in CLM-CN (Fig. 1). We add "(the rest is assumed to be SOM4, e.g., $f_{SOM4} = 1 - f_{LabileDOC} - f_{SOM1} ...$)"

**Comment 7**: P8L1-2: The turnover time of SOM3 and SOM4 are not listed – these fractions need to be better described or explain why they are excluded.

**Response 7**: add "(as the turnover time for SOM3 and SOM4 are 2 and 27 y, respectively, Fig. 1)".

**Comment 8**: P8L7-9: Nice explanation for "back of the envelop" biomass estimation

**Response 8**: Thanks.

**Comment 9**: P10L26-27: Are there other potential reasons why the rate of CO2 would stabilize? Limitation of some other resource? For instance, N? Does this study have evidence to support that rate of $CO_2$ respiration stabilized because of hydrolysis of polymers?

**Response 9**: We appreciated the reviewer for raising these questions. As we mention ahead of section 2.1, "While nitrogen (ammonium and nitrate) concentrations can affect carbon mineralization (Lavoie et al., 2011), we do not account for this effect because of a lack of nitrogen measurements from these experiments." As we mention earlier, we do not have specific direct evidence to support polymer hydrolysis as the limiting factor.

**Comment 10**: P11L8: parameter Fe3= 0.02 is above the max value in the range of observed values stated on P8L14, can the authors comment on why they might need to increase this value beyond observed values to help the model better match observations for Fe(II)? Do you have any suggestions for some other parameter that should be included or other parameter values that could be altered to help achieve a better model fit, while maintaining values within experimentally observed value range?

**Response 10**: The observed range is from another site. It is not directly applicable here. In the revision, we revise from "we start with $f_{fe3}$ = 0.005" to "While bioavailable Fe(III) in soils is not well defined (e.g., Hyacinthe et al. 2006; Poulton and Canfield 2005), we start with $f_{Fe3}$ = 0.005 and evaluate the impact with a range of values."

**Comment 11**: P11L11-14: How do these model observations relate to experimental data? Is there any experimental evidence (either from your original work or other soil Fe literature) to support that as Fe3 increases there is a decrease in CH4 resulting from competition between methanogens and iron reducers? Why wouldn't this also be the case when Fe3 = 0.01?

**Response 11**: We add "(rather than strict thermodynamic control, e.g., Bethke et al., 2011; direct inhibition, e.g., van Bodegom et al., 2004; or indirect inhibition through substrate competition, e.g., Mill et al., 2015, Reiche et al. 2008)". As discussed in these cited references, Fe reduction is known for inhibition of methanogenesis.

**Comment 12**: P11L29-31: This statement contradicts L25-26. L25-26 states as pH increases, CO2(aq) increases. L29-31 states as pH increases, CO2 (aq) decreases. Please provide an explanation.

**Response 12**: This was because CO2 in the aqueous phase here means a specific aqueous species rather than total CO2. To avoid this confusion, we add (aq) after CO2 and the sentence is revised from

"As the pH increases above the carbonic acid pKa (around 6.3 at standard condition), $CO_2(g)$ in the headspace and $CO_2$ in the aqueous phase decrease as $HCO_3^-$ becomes dominant, and the gas-phase fraction decreases dramatically." to

"As the pH increases above the carbonic acid pKa (around 6.3 at standard condition), $CO_2(g)$ in the headspace and $CO_{2(aq)}$ species decrease as $HCO_3^-$ becomes the dominant species in the aqueous phase, and the gas-phase fraction decreases dramatically."

**Comment 13**: P12L19: I keep having to look back at what "WEOC" means. I would recommend using some other terminology. Also, this sentence should reference Table 2 not Table 1.

**Response 13**: As suggested by the other two referees, we spell out WEOC. The table reference is corrected.

**Comment 14**: P12L20-22: Is this comparable? The values for rapid CO2 release in Figure 4 look nearly double or triple the observed values. It appears that CO2 values for organic center at a LabileDOC = 0.02 fit the experimental data best out of all of these scenarios.

**Response 14**: revise to "… the underprediction of the early $CO_2$ increase in the headspace are more or less mitigated."

**Comment 15**: P12L29: "high center polygon trough"? I thought "center" and "trough" were two different sampling sites? Please clarify and be consistent throughout the paper. Same error P13L6.

**Response 15**: revise from "…from the high center polygon trough" to "…from a trough location in a high center polygon…"

**Comment 16**: P13L19-20: I don't follow – how do these studies demonstrate that hydrolysis of macromolecular organics by extracellular enzymes is the rate limiting step? What about bioavailability? Limitation of some other resource?

**Response 16**: As we discuss earlier, we do not have direct unequivocal evidence for this.

**Comment 17**: P13L24-26: Please rewrite this sentence for clarity.

**Response 17**:  remove "(or produce substrate for)" and add "in the $s_{labile}$ = 0.2 case". The sentence reads:

"With $s_{labile}$ = 0.2, the model generally predicts less $CH_4$ and more $CO_2$ than the case with $s_{labile}$ = 0.4 because less SOM is assumed to respire through  the anaerobic pathway in the $s_{labile}$ = 0.2 case (Fig. S5)."

**Comment 18**: P13L31: "the model substantially underpredicts: : :" Please include a figure number.

**Response 18**: include figure number: Fig. 4b3.

**Comment 19**: P14L1: It could also be attributed to populations at that particular site grow more rapidly than the populations at other sites. Hard to say without a T0 measurement: : : I would tread lightly with this, you don't have strong experimental evidence to support this statement.

**Response 19**: remove ", indicating possible high initial abundance"

**Comment 20**: P14- first paragraph: The text says the opposite of what is demonstrated in Figure 5.

**Response 20**: The legend was wrong. It is corrected. The numbering for the subplots was moved to the right corner to avoid overlap with the legend.

**Comment 21**: Figure 5 shows the lower initial biomass results in more Ch4, FeII, pH increase, etc. Is it possible the figure legend is wrong?

**Response 21**: See Response 20

**Comment 22**: P14L10-12: OK, but if the OM soils are better buffered why are there rapid changes in pH for both the observed and experimental data for OM soils? FigureS6. OM soils appear to have rapid pH changes occur sooner than mineral soils, despite buffering? Please explain.

**Response 22**: The initial drastic drop in pH for OM soils are due to the fermentation of a large amount of initial labile carbon. Because of the abundance of simple substrates, Fe reduction and methanogenesis rates are high later, resulting in fast pH increase. It is really a complex nonlinear relationship.

**Comment 23**: P16L21: change "enhancing" to "enhances"

**Response 23**: revised.

**Comment 24**: P16: Transparent science! Thanks for making your code and data available!

**Response 24**: You are welcome. We are happy to share.

**Comment 25**: P17: It's unclear what a pH response and temperature response function are. Please better define. What is the reader supposed to take away from this information?

**Response 25**: add "(reaction rate adjustment factor as a function of pH)" and "(reaction rate adjustment factor as a function of temperature)". As we discuss in the introduction and results and discussion sections, the take-away is that these two response functions are an important source of uncertainty.

**Comment 26**: All tables and figures should be able to stand on their own. Improve caption text and add full legends (colors, symbols, and patterns defined in each figure).

**Response 26:** improved.

**Comment 27**: -Please format Table 2.

**Response 27**: this is reformed (see page 27).

**Comment 28**: -Figure 2 caption L5 add "as" after "such"

**Response 28**: added.

**Comment 29**: -Figure 5 caption text does not match figure. Legend suggests lowest initial biomass results in highest CH4. Please make full legend visible (partially covered up).

**Response 29**:  See Response 20.

[revised manuscript text omitted]

SOM4) are the major fractions of soil organic carbon but have a slow turnover time relative to the experiment duration, therefore, not shown.  See Fig.2 caption for more description about the model and experimental parameters.

[Figure]

**Figure S5: Impact of indirect respiration fraction ($s_{labile}$)  on predictions: less direct respiration means more simple substrates for iron reduction and methanogenesis . See Fig.2 caption for more description about the model and experimental parameters.**

[Figure]

**Figure S6: Impact of specified organic matter in WHAM on predictions: more organic matter means more pH buffer.** See Fig.2 caption for more description about the model and experimental parameters.

[Figure]

**Figure S7: Comparison of the impact of different pH response functions (CLM4Me, TEM, and DLEM) on predictions. pH response function can be a substantial source of prediction uncertainty. See Fig.2 caption for more description about the model and experimental parameters.**

[Figure]

**Figure S8: Fig. 7 with arithmetic vertical scale.**

[Figure]

**Figure S9: Comparison of impact of different temperature response functions (CLM-CN, CENTURY, Ratkowsky Equation with $T_m$ =260) on predictions.** Predictions are sensitive to temperature response function, which can introduce large prediction uncertainty. See Fig.2 caption for more description about the model and experimental parameters.

[Figure]

**Figure S10: Comparison of impact of different temperature response functions (CLM-CN, Arrhenius equation ($E_a$), $Q_{10}$ Equation) on predictions. Predictions are sensitive to temperature response function, which can introduce large prediction uncertainty. See Fig.2 caption for more description about the model and experimental parameters.**

[Figure]

**Figure S11: Impact of headspace volume on predictions: increase in headspace volume results in decrease in headspace (f1-6) and aqueous (g1-6) $CO_2$ concentration, slower pH increase and biogeochemical reaction rates, and generally less $CO_2$ and $CH_4$ production prediction. As an exception, predicted $CO_2$ production is increases with increasing headspace volume for the center oganic soils. The impact is not linear as the underlying biogeochemical processes are nonlinear. TOTC = initial total organic carbon. Ac = organic acids as acetate. See Fig.2 caption for more description about the model and experimental parameters.**